# ACON: Optimizing Context Compression for Long-horizon LLM Agents

**Minki Kang** [1] [*]   **Wei-Ning Chen** [2]   **Dongge Han** [2]   **Huseyin A. Inan** [2]   **Lukas Wutschitz** [2]   **Yanzhi Chen** [2] [3]
**Robert Sim** [2]   **Saravan Rajmohan** [2]

## Abstract

Large language models (LLMs) are increasingly deployed as agents in dynamic real-world environments, where success depends on maintaining precise records of actions and observations. However, the resulting unbounded context growth in long-horizon agentic tasks makes two critical bottlenecks: prohibitive inference memory costs and reasoning degradation due to irrelevant information. Existing compression methods fail to fully address this, often relying on brittle heuristics or requiring parameter updates impractical for proprietary or large-scale LLMs. We introduce **Agent Context Optimization** (ACON), a unified framework that optimally compresses both observations and history into concise, informative representations. Distinct from prior works, ACON employs an optimization in natural language space: it iteratively refines compression guidelines based on failure analysis of the agent, ensuring critical state information is preserved without model fine-tuning. To further minimize computational overhead, we distill the optimized compressor into smaller models. Experiments on AppWorld, OfficeBench, and Multi-objective QA demonstrate that ACON reduces peak token usage by 26–54% while improving task success over existing compression baselines. Notably, it enables smaller LMs to function effectively as long-horizon agents, achieving up to 46% performance improvement by mitigating context distraction. Our code is available at https://github.com/microsoft/acon.

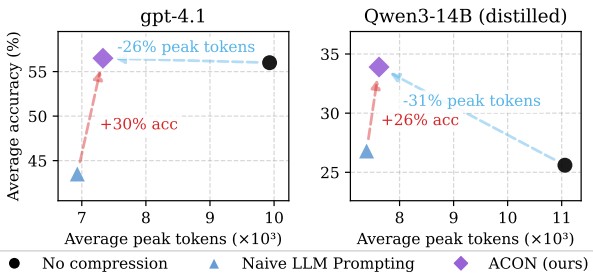

*Figure 1.* **Accuracy-Peak tokens trade-off** on AppWorld benchmark (Trivedi et al., 2024). We compare average accuracy versus peak tokens in history compression. ACON (ours) reduces token cost while preserving accuracy for the large model (gpt-4.1) relative to a naive prompting baseline, and even improves accuracy on smaller model (Qwen-14B). More results are in Section 4.

## 1. Introduction

Large language models (LLMs) have become the backbone of AI agents, enabling them to plan and act in dynamic environments (Yao et al., 2023). However, these tasks often unfold over extended horizons, requiring the agent to maintain a continuous record of observations, tool outputs, and evolving states. In such settings, context is not auxiliary but foundational; losing a single detail such as a file path or an API parameter can derail the entire workflow. As interactions accumulate, context grows unbounded as shown in Figure 2, making two major bottlenecks. First, the inference memory cost of transformers scales with context length, resulting long-horizon reasoning computationally prohibitive due to the massive KV cache requirements (Vaswani et al., 2017). Second, excessively long contexts dilute relevant information, distracting the model with outdated or extraneous details and degrading decision quality (Shi et al., 2023).

The challenge of managing this context is particularly critical in productivity scenarios, such as email management or workflow automation, where agents must coordinate across heterogeneous tools (Trivedi et al., 2024; Wang et al., 2024b). Unlike simpler conversational tasks, these environments demand the preservation of diverse signal types: factual history, action-outcome relationships, success preconditions, and future decision cues. Naive strategies like token truncation or generic summarization are insufficient, as they easily discard these critical details essential for multi-

---

[*]Work done during the internship at Microsoft.   [1]KAIST [2]Microsoft [3]University of Cambridge. Correspondence to: Minki Kang <minkikang@kaist.ac.kr>.

*Proceedings of the 43rd International Conference on Machine Learning, Seoul, South Korea. PMLR 306, 2026. Copyright 2026 by the author(s).*

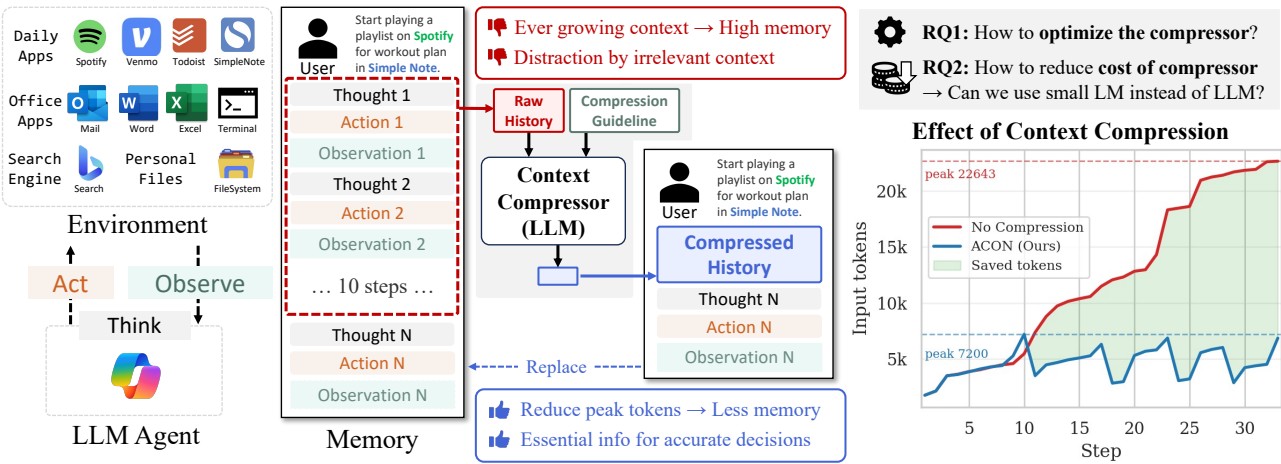

*Figure 2.* **Motivation: Unbounded context in LLM agents.** Continuous interactions lead to ever-growing contexts that incur high memory usage (red line). This motivates the need for compression, yet raises key questions on *how to optimize the compressor* and *reduce its cost*. We address these with ACON through a **compression guideline optimization** and **compressor distillation**, effectively reducing peak tokens (blue line) while preserving essential information.

step reasoning. Consequently, effective compression must balance aggressive reduction with the precise retention of task-relevant state information.

Existing compression approaches, however, fail to fully address these agent-specific needs. Dialogue-oriented systems (Packer et al., 2023) focus on conversational coherence rather than state tracking, while document-centric methods (Jiang et al., 2024) assume single-step reasoning where context can be discarded after use. While recent agent-focused methods attempt to bridge this gap, they face significant limitations: heuristic-based approaches (Deng et al., 2023; Smith, 2025) are often brittle and narrowly specialized, limiting their robustness. Meanwhile, model optimization-based approaches (Zhou et al., 2025; Sun et al., 2025) typically entangle compression with the agent model, making them difficult to apply directly to proprietary, API-based LLMs, where gradient-based updates to the underlying model are infeasible.

To address these challenges, we introduce **Agent Context Optimization (ACON)**, a unified framework for optimizing the compression of both environment observations and interaction histories. Distinct from heuristic approaches that rely on static rules and model optimization-based methods that require model parameter updates, ACON introduces a **compression guideline optimization** directly in natural language space. This method refines compressor prompts through failure analysis ensuring that critical environment-specific signals are preserved after compression without altering the agent model weight (Pryzant et al., 2023; Yang et al., 2024a; Yüksekgönül et al., 2025; Han et al., 2025). It makes ACON purely model-agnostic and directly applicable to agents based on proprietary, API-based LLMs.

ACON yields three key advantages over previous works. First, the guideline optimization enables environment-specific compression rules to be derived consistently across diverse agentic tasks, overcoming the brittleness of hand-crafted heuristics. Second, by retaining essential information, optimally compressed contexts not only reduce memory costs but also improve decision quality, allowing smaller models to act more effectively by mitigating distraction. Third, we validate that these optimized compressors can be distilled into smaller models, demonstrating that the compression module itself can be deployed with minimal computational overhead.

We validate ACON on three multi-step agent benchmarks: AppWorld (Trivedi et al., 2024), OfficeBench (Wang et al., 2024b), and Multi-objective QA (Kwiatkowski et al., 2019; Zhou et al., 2025), each requiring 15+ interaction steps. Our empirical results demonstrate clear advantages of ACON: (1) lowers peak token usage of agents by 26–54% while improving task success compared to existing compression baselines (2) enables effective distillation of the context compressor into smaller models, preserving 95% of the teacher's accuracy, thereby reducing the overhead of the compression (3) allows small LMs to function more effectively as agents, improving performance by 32% on AppWorld, 20% on OfficeBench, and 46% on Multi-objective QA by mitigating the distraction of long contexts. Our result highlights on AppWorld benchmark are in Figure 1.

In summary, our work makes the following contributions:

- We propose **Agent Context Optimization** (ACON), a framework for optimizing compression of both environment observations and interaction histories, tailored to

multi-step, long-horizon agentic tasks.

- We develop a failure-driven compression guideline optimization. This approach is model-agnostic, making it readily applicable to any LLM, including proprietary API-based models, without requiring weight updates.

- We enable cost-efficient deployment of optimized compressors by distilling them into smaller models, preserving over **95%** of the teacher's performance while reducing the overhead of compression.

- We validate ACON on AppWorld, OfficeBench, and Multi-objective QA, showing that it reduces peak token usage by **26–54%** while improving task success over existing compression baselines with LLMs, and enabling small LMs to achieve **20–46%** performance improvements.

## 2. Related Works

**Long-horizon LLM agents.** Large language model (LLM) agents extend pretrained models beyond static single-step reasoning tasks (e.g., RAG-based QA, math problem solving, or code generation) to interactive decision-making in dynamic environments (Yao et al., 2023; Wang et al., 2024a; Team et al., 2025; OpenAI, 2025a). Unlike chatbots or solvers that return an answer in one pass, agents must iteratively observe their surroundings, select tools, and execute actions while revising their plans based on feedback (Shridhar et al., 2021; Jimenez et al., 2024; Zhou et al., 2024; Wei et al., 2025; Xie et al., 2024; Bonatti et al., 2024). Recent work highlights the importance of *long-horizon LLM agents*, which tackle tasks that unfold over dozens to hundreds of steps and require coordination across multiple applications and tools (Kwa et al., 2025; Trivedi et al., 2024; Wang et al., 2024b; 2025b). A central challenge in these scenarios lies in managing the *dynamic long context*, where the agent must retain multi-step interaction histories and handle diverse observations produced by heterogeneous environments.

**Context compression for LLMs.** Managing this ever-growing context has been a longstanding challenge, and a variety of approaches have been proposed to compress LLM inputs. Prior works on context compression can be broadly grouped into three directions: document- or retrieval-based compression (Seo et al., 2025; Li et al., 2023; Xu et al., 2024; Yoon et al., 2024; Zhou et al., 2025; Jiang et al., 2024; Shandilya et al., 2025), dialogue memory summarization (Xu et al., 2025; Maharana et al., 2024; Wang et al., 2025a), and low-level KV cache compression (Zhang et al., 2025). While each line of research has demonstrated benefits in its respective setting, they remain insufficient for the dynamic and heterogeneous contexts required by long-horizon agents, where the relevance of in-

formation frequently shifts as the agent progresses.

Beyond general compression, several recent works have explored context compression specifically for LLM agents (Deng et al., 2023; Lee et al., 2025; Smith, 2025; Yu et al., 2025). However, these approaches either rely on naive prompting or target narrow domains, limiting their broader applicability. Another related line of works treats context compression as an agent action (Zhou et al., 2025; Ye et al., 2025; Sun et al., 2025), employing reinforcement learning to optimize the model for both compression and agent action policy. However, such methods inherently update the model to couple reasoning with compression, typically requiring access to internal weights. In addition, Re-Sum (Wu et al., 2026) shares the motivation of extending long-horizon agents through summarized contexts, but it optimizes the policy model to better utilize summaries.

**In contrast,** we introduce a universal optimization framework for agent context compression that is applicable to any arbitrary LLMs. Our framework distinguishes itself by supporting both history and observation compression and providing a generalizable optimization process for the compression. Since our approach is entirely **model-agnostic**, it remains equally effective for both open-source models and proprietary API-based LLMs. A detailed analysis is provided in Appendix C.

## 3. Agent Context Optimization (ACON)

We present Agent Context Optimization (ACON), a unified framework for optimized history and observation compression in long-horizon LLM agents. We begin by formulating the agent task and defining context cost in Section 3.1. Next, in Section 3.2, we introduce generative compression with LLMs for both history and observation, and formalize the associated optimization objective and its challenges. We then propose our optimization method in Section 3.3, followed by a distillation that enables smaller models for compressions to reduce the compression cost (Section 3.4).

### 3.1. Problem Formulation

**Task.** An agentic task is formulated as a Partially Observable Markov Decision Process (POMDP) $\mathcal{E} = \langle \mathcal{S}, \mathcal{A}, \mathcal{O}, \mathcal{T}, \mathcal{R} \rangle$ with state space $\mathcal{S}$, action space $\mathcal{A}$, and observation space $\mathcal{O}$. The transition function $\mathcal{T}(s, a) \rightarrow (s', o)$ is deterministic, and it is determined by the environment. Specifically, it executes an action $a \in \mathcal{A}$ in the environment and returns the next state and observation. The reward function $\mathcal{R}$ returns the reward given the terminal state $s_T$. The terminal state is arrived when the transition function receives the special action (e.g., finish_task).

An LLM agent interacts with the environment to get information for making a decision to achieve a given task

$o_0$ through multiple steps. For each step $t$, the LLM $\mathcal{M}$ generates the action $a_t$ followed by its reasoning at each step (Yao et al., 2023; Wang et al., 2024a) given the inter-action history $\boldsymbol{h}_{t-1} = (o_0, a_0, o_1, a_1, \cdots, o_{t-1}, a_{t-1})$ and the latest observation $o_t$:

$$\mathcal{M}(o_t, \boldsymbol{h}_{t-1}; \theta, \mathcal{P}_{\mathsf{agent}}) \mapsto a_t, \qquad (1)$$

where $\theta$ refers to the pre-trained parameters of the LLM and $\mathcal{P}_{\mathsf{agent}}$ is the prompt that consists of a general environment description, tools, output format, and few-shot examples in natural language.

**Cost function for context.** We assume that the LLM agent's parameters $\theta$ and the task and system prompt $\mathcal{P}_{\mathsf{agent}}$ are fixed. We define a cost function $\mathcal{C}$ that measures the cost of encoding the dynamic context during action generation at each step such as $\mathcal{O}(n)$ computational cost of a transformer for decoding given $n$ input tokens. The cost function takes the interaction history $\boldsymbol{h}_{t-1}$, and the latest observation $o_t$ as input and returns the per-step cost:

$$C(\boldsymbol{H}) = \sum_{t=1}^{T} \mathcal{C}(\boldsymbol{h}_{t-1}, o_t), \qquad (2)$$

where $C$ is the total cost of completing the task, $\boldsymbol{H} = \{\boldsymbol{h}_{t-1}, o_t\}_{t=1}^{T}$ denotes the sequence of history and observation of each step. Typically, $C$ is proportional to the summation of token lengths of action and observations in each step, $\boldsymbol{h}_{t-1}$ and $o_t$. While the cost of system prompt is static, the costs from interaction histories are **unbounded**, leaving the user with only two options: terminate the task early or truncate the context heuristically to a maximum length. This raises the central question: *how can we compress context more effectively than such heuristics?*

### 3.2. History & Observation Compression with LLMs

To address this challenge, we use an LLM $f(\cdot; \phi, \mathcal{P})$, parameterized by pre-trained weights $\phi$ and a compression guideline $\mathcal{P}$, to minimize context cost defined in Equation 2 (e.g., *summarize the given interaction history*). As in Equation 1, the LLM receives two inputs at each step: the interaction history $\boldsymbol{h}_{t-1}$ and the latest observation $o_t$. This introduces two options for context compression:

**History compression.** The interaction history accumulates both environment observations and agent actions. In long-horizon tasks, this history can grow excessively large. To manage its length, we apply history compression only when the length exceeds a predefined threshold $T_{\mathsf{hist}}$:

$$\boldsymbol{h}'_t = f(\boldsymbol{h}_t; \phi, \mathcal{P}_{\mathsf{hist}}) \text{ if } |\boldsymbol{h}_t| > T_{\mathsf{hist}}, \quad \boldsymbol{h}_t \text{ otherwise.} \qquad (3)$$

The compressed history $\boldsymbol{h}'_t$ replaces the raw history in Equation 1. This selective compression ensures that the

overhead of invoking the compressor is incurred only when necessary (Smith, 2025).

**Latest observation compression.** Given an action $a$, the environment returns an observation $o$ according to the transition function $\mathcal{T}(s, a) \to (s', o)$. We similarly apply observation compression only when the observation length exceeds a threshold $T_{\mathsf{obs}}$:

$$o'_t = f(o_t, \boldsymbol{h}_{t-1}; \phi, \mathcal{P}_{\mathsf{obs}}) \text{ if } |o_t| > T_{\mathsf{obs}}, \quad o_t \text{ otherwise.} \qquad (4)$$

This mechanism avoids unnecessary overhead when $o_t$ is already short, while still reducing redundant or distracting content in long observations (Xu et al., 2024; Deng et al., 2023; Lee et al., 2025). The compressed one $o'_t$ replaces the raw one in Equation 1 and is stored in the interaction history $\boldsymbol{h}$.

In both cases, the compressor LLM selects information to preserve based on its learned prior knowledge of importance. However, there is **no guarantee** that the salient details required for successful task completion are retained. The agent context effectively serves as a **world model of the environment**, encompassing diverse forms of information such as causal relations (*e.g.,* email leaves drafts), evolving states (*e.g.,* account balance), preconditions (*e.g.,* login required), and task-relevant decision cues (*e.g.,* due dates). Effective context compression must therefore accommodate this heterogeneous and dynamic nature of agent context, ensuring that the most critical signals are preserved for long-horizon reasoning and task success.

**Optimization objective.** We optimize the compressor parameters $\psi \triangleq (\phi, \mathcal{P})$ to maximize task reward while minimizing context cost. At each step $t$, the compressor produces either a compressed history $\boldsymbol{h}'_t = f_{\mathsf{hist}}(\boldsymbol{h}_t; \psi)$ or observation $o'_t = f_{\mathsf{obs}}(o_t, \boldsymbol{h}_{t-1}; \psi)$. Let the compressed context be

$$\boldsymbol{H}'(\psi) = \{\boldsymbol{h}'_{t-1}, o'_t\}_{t=1}^{T}, \qquad (5)$$

$$C(\boldsymbol{H}'(\psi)) = \sum_{t=1}^{T} \mathcal{C}(\boldsymbol{h}'_{t-1}, o'_t). \qquad (6)$$

With the agent $\mathcal{M}(\cdot; \theta, \mathcal{P}_{\mathsf{agent}})$ fixed, the environment induces a trajectory $\tau(\psi)$ and terminal state $s_T(\psi)$ when the agent conditions on $\boldsymbol{H}'(\psi)$. Our learning objective is

$$\max_{\psi} \underbrace{\mathbb{E}\big[\mathcal{R}\big(s_T(\psi)\big)\big]}_{\text{maximize}} - \lambda \underbrace{\mathbb{E}\big[C\big(\boldsymbol{H}'(\psi)\big)\big]}_{\text{minimize}}, \quad \lambda \geq 0, \quad (7)$$

where $\lambda$ is a multiplier and the expectations are over tasks.

**Challenges.** The optimization objective in Equation 7 is difficult to optimize in practice because there is no gold supervision for compression, the reward is sparse and only revealed at the end of the trajectory, and the context cost

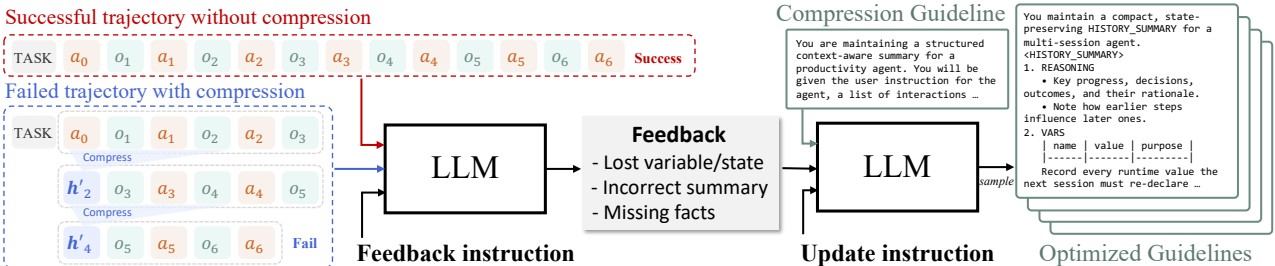

*Figure 3.* **Compression Guideline Optimization.** Feedback is generated by contrasting successful trajectories (no compression) with failed ones (with compression). The collected feedback is then used by LLM to refine the compression guidelines.

is defined over discrete quantities, which precludes direct gradient computation. While these properties naturally motivate reinforcement learning (RL) (Sutton & Barto, 2018), applying RL introduces additional obstacles: (1) updating the parameters $\phi$ of a LLM with RL can be computationally prohibitive, (2) environment rollouts are extremely expensive since each reward requires multi-step executions of both agent and compressor, and (3) policy gradient estimates suffer from high variance since compression quality is only indirectly evaluated through eventual task success.

### 3.3. Optimizing Compression Guidelines

To overcome these challenges, we propose to optimize **compression guidelines** $\mathcal{P}$ (natural language prompts) for context compression, rather than fine-tuning model parameters $\phi$. Trajectories under compressed contexts provide *dense signals* about the quality of compression. For example, if the agent fails with compressed context while succeeding without compression, this indicates that the compressed context may have lost crucial information. Such trajectory-level comparisons yield richer feedback than scalar rewards (e.g., binary task success).

We instantiate this idea as prompt optimization using an LLM as the optimizer, where the natural language prompt $\mathcal{P}$ is refined via feedback expressed in natural language (Yang et al., 2024a; Yüksekgönül et al., 2025; Khattab et al., 2024; Choi et al., 2026). We introduce **compression guideline optimization** based on *contrastive task feedback*.

On the training set $\mathcal{D}_{\text{train}}$, we run the LLM agent both without and with context compression to obtain baseline context $\boldsymbol{H}$ and compressed context $\boldsymbol{H}'$. We collect tasks where the agent succeeds with $\boldsymbol{H}$ but fails with $\boldsymbol{H}'$, forming a contrastive subset $\mathcal{D}_{\text{cont}}$. For each task in $\mathcal{D}_{\text{cont}}$, we query a optimizer LLM with the context before and after compression to obtain natural language feedback:

$$\text{Feedback} = \text{LLM}(\text{Feedback Instruction}, \boldsymbol{H}, \boldsymbol{H}'). \quad (8)$$

This feedback serves as a natural language gradient (Yüksekgönül et al., 2025), indicating how the compression guideline $\mathcal{P}$ should be refined. We then aggregate feedback

from multiple trajectories:

$$\mathcal{P}^{(1)} = \text{LLM}(\text{Update Inst.}, \mathcal{P}^{(0)}, \|_{i=1}^{n}\text{Feedback}_i), \quad (9)$$

where $\|$ is concatenation of feedbacks from each task, which corresponds to a batch optimization step in textual gradient descent (Yüksekgönül et al., 2025). We also generate multiple candidate prompts $\{\mathcal{P}_k^{(1)}\}$, evaluate them on $\mathcal{D}_{\text{cont}}$, and select the best-performing one. We refer this process as *utility maximization* step $\overline{\text{UT}}$ as it primarily maximizes the first term (task reward) of Equation 7.

However, optimizing only for reward may neglect the context cost (second term in Equation 7). To address this, motivated by alternating optimization, we perform a second iteration that conditions only on successful task with compressed context, asking the LLM to generate feedback about which information was actually used during execution. This refines $\mathcal{P}^{(1)} \to \mathcal{P}^{(2)}$, encouraging shorter yet sufficient contexts. We refer this additional process as *compression maximization* step $\overline{\text{CO}}$ as it minimizes the second term (context cost) of Equation 7.

We illustrate overall process in Figure 3. Algorithm 1 and prompts are in Appendix B.

### 3.4. Distilling Context Compression into Small Models

While compression guideline optimization enables effective compression, repeatedly invoking the large LLM for compression adds substantial overhead. To reduce this cost, we **distill the compressor into a smaller model**. The teacher with optimized guideline $\mathcal{P}^*$ (parameters $\phi_{\text{T}}$) generates compressed outputs $\boldsymbol{y}$ from input $\boldsymbol{x}$, which supervise the student (parameters $\phi_{\text{S}}$). We train the student with a cross-entropy objective (Kim & Rush, 2016) with input-output pair $(\boldsymbol{x}, \boldsymbol{y})$, where $(\boldsymbol{x}, \boldsymbol{y}) = (\boldsymbol{h}_t, \boldsymbol{h}'_t)$ for Equation 3 or $(\boldsymbol{x}, \boldsymbol{y}) = ((\boldsymbol{h}_{t-1}, o_t), o'_t)$ for Equation 4:

$$\min_{\phi_{\text{S}}} \mathbb{E}_{(\boldsymbol{x}, \boldsymbol{y}) \sim \mathcal{D}_{\text{train}}^+} \left[ -\sum_{n=1}^{L_{\boldsymbol{y}}} \log f(\boldsymbol{y}_n \mid \boldsymbol{x}, \boldsymbol{y}_{<n}; \phi_{\text{S}}, \mathcal{P}^*) \right], \quad (10)$$

where $\mathcal{D}_{\text{train}}^+$ denotes tasks where the teacher succeeds with compressed context.

*Table 1.* Results across different difficulty levels on **Appworld** benchmark (test-normal). Each block reports accuracy (task goal completion score), steps, peak input tokens ($10^3$), and dependency ($10^6$) for agents. Best results in each column are highlighted in bold. Rows in blue background indicate the results from ours. ACON consistently improves accuracy while reducing peak tokens and dependency, with ACON $\overline{\text{UT}}\overline{\text{CO}}$ achieving the best overall performance.

| Method | Average (168) | | | | Easy (57) | | | Medium (48) | | | Hard (63) | | |
|---|---|---|---|---|---|---|---|---|---|---|---|---|---|
| | Acc. ↑ | Steps ↓ | Peak ↓ | Dep.↓ | Acc. ↑ | Peak ↓ | Dep.↓ | Acc. ↑ | Peak ↓ | Dep.↓ | Acc. ↑ | Peak ↓ | Dep.↓ |
| **Agent:** `gpt-4.1` / **Compressor:** `gpt-4.1` | | | | | | | | | | | | | |
| No compression | 56.0 | **16.14** | 9.93 | 5.96 | 80.7 | 7.57 | 2.98 | 47.9 | 10.10 | 5.36 | **39.7** | 11.95 | 9.11 |
| **History Compression** | | | | | | | | | | | | | |
| FIFO | 45.8 | 28.48 | 6.73 | 5.69 | 84.2 | 5.85 | 2.89 | 39.6 | 7.26 | 6.24 | 15.9 | **7.14** | 7.80 |
| Retrieval | 27.4 | 33.17 | 8.39 | 6.68 | 61.4 | 7.40 | 3.97 | 12.5 | 8.74 | 7.72 | 7.9 | 9.02 | 8.33 |
| LLMLingua | 39.3 | 24.42 | 7.50 | 6.37 | 66.7 | 6.38 | 3.04 | 37.5 | 8.04 | 7.39 | 15.9 | 8.09 | 8.59 |
| Prompting | 43.5 | 24.01 | 6.93 | 5.29 | 66.7 | 6.36 | 2.84 | 41.7 | 7.10 | 5.36 | 23.8 | 7.31 | 7.48 |
| ACON $\overline{\text{UT}}$ | 51.2 | 20.92 | 7.17 | 4.49 | 77.2 | 6.45 | 2.43 | 50.0 | 7.39 | 4.47 | 28.6 | 7.65 | **6.37** |
| ACON $\overline{\text{UT}}\overline{\text{CO}}$ | **56.5** | 22.82 | 7.33 | 4.69 | **86.0** | 7.09 | 2.84 | **56.2** | 7.48 | 4.43 | 30.2 | 7.44 | 6.55 |
| **Observation Compression** | | | | | | | | | | | | | |
| LLMLingua | 32.1 | 18.16 | 8.17 | 6.01 | 54.4 | 5.78 | 2.33 | 29.2 | 8.24 | 5.23 | 14.3 | 10.29 | 9.92 |
| Prompting | 42.3 | 17.38 | **6.58** | **4.09** | 64.9 | **4.92** | **1.88** | 35.4 | **6.96** | **4.11** | 27.0 | 7.79 | 6.07 |
| ACON $\overline{\text{UT}}$ | 47.0 | 16.67 | 7.62 | 5.08 | 70.2 | 5.87 | 2.21 | 45.8 | 7.79 | 5.00 | 27.0 | 9.07 | 7.73 |
| ACON $\overline{\text{UT}}\overline{\text{CO}}$ | 53.6 | 18.12 | 7.43 | 4.93 | 82.5 | 5.66 | 2.63 | 47.9 | 7.30 | 4.43 | 31.8 | 9.14 | 7.50 |

Once trained, the student replaces the teacher during inference, decoupling decision making from compression. This two-stage pipeline, guideline optimization then distillation, achieves effective compression with a much smaller model ($|\phi_{\mathsf{T}}| \gg |\phi_{\mathsf{S}}|$):

$$f(\cdot; \phi_{\mathsf{T}}, \mathcal{P}) \xrightarrow{\text{prompt optim.}} f(\cdot; \phi_{\mathsf{T}}, \mathcal{P}^*) \xrightarrow{\text{distill}} f(\cdot; \phi_{\mathsf{S}}, \mathcal{P}^*). \quad (11)$$

## 4. Experiments

We evaluate ACON on three challenging benchmarks that require multi-step interactions across diverse domains. Our experiments are designed to address the following key questions:

- How well does ACON improve token efficiency while preserving performance? (Section 4.2)
- Does distilling the compressor reduce its size while maintaining agent performance? (Section 4.3)
- Can ACON help small, distilled LM agents perform better under long contexts? (Section 4.4)

### 4.1. Experimental Setup

**Benchmarks & Metrics.** We focus on long-horizon agentic task benchmarks that require 10+ interaction steps on average: **(1) AppWorld** (Trivedi et al., 2024): Main benchmark with 9 simulated apps (e.g., Venmo, Spotify, SimpleNote) and ∼100 simulated users. Performance is measured by task completion score. **(2) OfficeBench** (Wang et al., 2024b): Productivity tasks across 6 apps (e.g., Word, Excel, Email), operating

on simulated documents. Performance is measured by benchmark-defined accuracy functions. **(3) 8-objective QA** (Kwiatkowski et al., 2019; Zhou et al., 2025): QA benchmark where agents interact with a search tool to answer 8 questions and output a consolidated answer set. Performance is the average of Exact Match (EM) and F1 scores across 8 questions.

In addition to task-specific performance metrics, we report three token efficiency metrics following prior work (Zhang et al., 2025; Zhou et al., 2025): **(1) Steps:** The average number of interaction steps per task. **(2) Peak Tokens:** The maximum context length encountered across all steps. **(3) Dependency:** The cumulative dependency of each generated action on prior tokens, measuring how much generation relies on the context history. Full details are provided in the Appendix B.

Throughout all experiments, we use ReAct agent (Yao et al., 2023). For the detailed tool format, we follow the convention of each benchmark.

**Baselines.** **(1) No Compression:** full uncompressed context. **(2) FIFO:** keep the most recent $k$ interactions, discarding earlier ones (Yang et al., 2024b). **(3) Retrieval:** select $k$ past interactions most similar to the current query via embedding search (Xu et al., 2025). **(4) LLMLingua:** extractive compression with an encoder-only LM (Jiang et al., 2023; Pan et al., 2024). **(5) Prompting:** naive baseline using a general compression instruction (Smith, 2025; Lee et al., 2025).

**Our Methods.** We evaluate two versions of ACON. **(1) ACON $\overline{\text{UT}}$** utilizes an *optimized guideline* for context com-

*Table 2.* Results on **OfficeBench** and **8-objective QA** benchmarks. We report performance metrics (acc/EM/F1) along with steps, peak input tokens ($10^3$), and dependency ($10^6$). Best values are in **bold**. Rows in blue are $\boxed{\text{ours}}$. ACON consistently improves accuracy/efficiency trade-offs.

*(a)* OfficeBench

| Method | Acc. ↑ | Steps ↓ | Peak ↓ | Dep. ↓ |
|---|---|---|---|---|
| **Agent: gpt-4.1 / Compressor: gpt-4.1** | | | | |
| No Compression | **76.84** | 11.52 | 7.27 | 4.43 |
| **History Compression** | | | | |
| FIFO | 67.37 | 12.26 | **4.02** | 2.64 |
| Retrieval | 65.26 | 16.20 | 4.33 | 2.06 |
| LLMLingua | 70.53 | 10.89 | 4.65 | 1.85 |
| Prompting | 71.58 | **10.13** | 4.40 | **1.10** |
| ACON UT | 74.74 | 13.13 | 4.93 | 3.85 |
| ACON UTCO | 72.63 | 11.54 | 4.54 | 1.91 |
| **Observation Compression** | | | | |
| LLMLingua | 71.58 | 11.89 | 7.38 | 6.14 |
| Prompting | 55.79 | 12.24 | 6.44 | 2.68 |
| ACON UT | 73.68 | 10.83 | 6.55 | 3.85 |
| ACON UTCO | 72.63 | 10.28 | 6.17 | 2.88 |

*(b)* 8-objective QA

| Method | EM ↑ | F1 ↑ | Steps ↓ | Peak ↓ | Dep. ↓ |
|---|---|---|---|---|---|
| **Agent: gpt-4.1 / Compressor: gpt-4.1** | | | | | |
| No compression | 0.366 | 0.488 | 15.78 | 10.35 | 3.32 |
| **History Compression** | | | | | |
| FIFO | 0.293 | 0.388 | 19.26 | 5.09 | 2.51 |
| Retrieval | 0.331 | 0.438 | 20.06 | 5.11 | 2.62 |
| LLMLingua | 0.363 | 0.481 | 17.68 | 5.68 | 2.24 |
| Prompting | **0.376** | 0.478 | 18.70 | 4.73 | 1.66 |
| ACON UT | 0.373 | **0.494** | 17.14 | 4.71 | 1.57 |
| ACON UTCO | 0.335 | 0.458 | 17.79 | **4.65** | **1.50** |
| **Observation Compression** | | | | | |
| LLMLingua | 0.320 | 0.414 | 14.23 | 5.16 | 1.35 |
| Prompting | 0.288 | 0.397 | **11.64** | **3.41** | **0.45** |
| ACON UT | 0.364 | 0.475 | 16.33 | 4.97 | 1.28 |
| ACON UTCO | 0.336 | 0.461 | 14.00 | 4.22 | 0.81 |

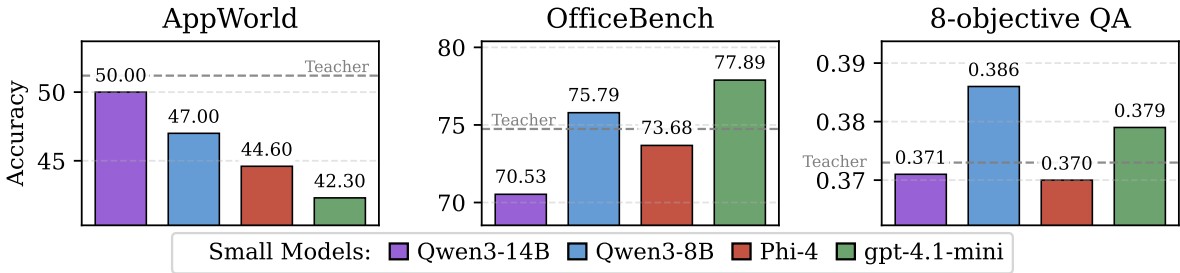

*Figure 4.* **Results of distilled compressors on history compression** with gpt-4.1 as the agent. Grey dotted lines denote performance using the gpt-4.1 teacher compressor. Student models (Qwen3-14B, Qwen3-8B, Phi-4) are distilled from gpt-4.1 compressor using the optimized compression guideline after UT step, and evaluated across all benchmarks. We also include gpt-4.1-mini without distillation, showing that even a small model can serve as an effective compressor without additional training.

pression after utility maximization step. **(2)** ACON UTCO applies compression maximization CO after utility maximization UT, aiming for shorter but informative compression. We apply only single step per each step for experiments. Additional analysis on the number of steps is in Appendix C.

### 4.2. Overall performance and token efficiency

In Table 1 and Table 2, we first evaluate ACON using on gpt-4.1 (OpenAI, 2025b) for both agent and compressor, which already achieves strong results on benchmarks.

For history compression, as shown in Table 1, on App-World, **ACON reduces peak tokens by over 25%** while preserving the accuracy of the no compression upper bound, outperforming all baselines that suffer severe degradation on medium and hard tasks spanning longer steps. On OfficeBench (Table 2a), ACON lowers peak context size by

nearly 30% while maintaining accuracy above 74%. On 8-objective QA (Table 2b), ACON even surpasses the no compression baseline in EM/F1 while reducing peak tokens and dependency by 54.5% and 61.5%, respectively. For observation compression, ACON consistently outperforms all baselines confirming that compression guideline optimization is effective for compressing not only history but also raw observations.

Applying only the utility maximization step (UT) improves performance while reducing token cost across all benchmarks, whereas the compression maximization step (CO) further lowers token cost but may slightly hurt accuracy depending on the environment. This trade-off suggests a practical guideline for choosing between the two variants. In verbose and noisy environments such as App-World, where observations contain redundant API outputs and distractors, UTCO is often preferable because aggressive pruning can improve both efficiency and focus. In

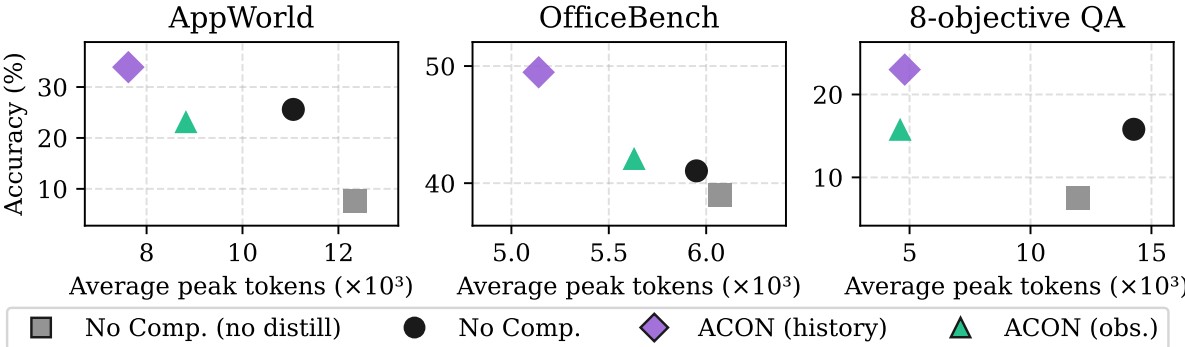

*Figure 5.* **Performance-efficiency trade-off of the Qwen3-14B agent** distilled from gpt-4.1 trajectories. For distilled compressors, we use the same distillation setting as in Figure 4. Compared to the baseline without compression, our framework ACON provides compressed trajectories combined with a distilled compressor, enabling the distilled agent to achieve consistently higher accuracy while requiring substantially fewer peak input tokens across all benchmarks.

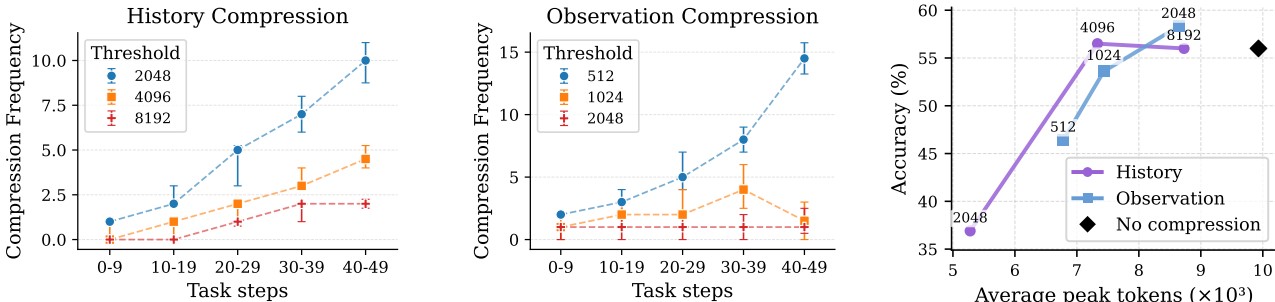

*Figure 6.* **Ablation studies on thresholds for compression** on AppWorld with gpt-4.1. (1) the number of compressions (compression frequency) for each length of task trajectories (task steps). (2) the performance comparison for each threshold setting.

contrast, in high-fidelity information-seeking tasks such as OfficeBench and 8-objective QA, $\overline{\text{UT}}$ is generally safer because over-compression may remove subtle facts needed for final decisions. Therefore, $\overline{\text{UTCO}}$ is recommended when observations contain substantial redundancy, while $\overline{\text{UT}}$ is preferable when precise fact retention is critical.

### 4.3. Compressor distillation

We distill the compressor with optimized guidelines after $\overline{\text{UT}}$ step into smaller models such as Qwen3-14B, Qwen3-8B (Yang et al., 2025), and Phi-4 (Abdin et al., 2024) using LoRA (Hu et al., 2022). As shown in Figure 4, the **distilled compressors retain over 95% of the performance of the gpt-4.1 teacher** (indicated by the grey dotted line) while reducing computational overhead. We also observe that gpt-4.1-mini, even without any distillation, can serve as an effective lightweight compressor on OfficeBench and QA. This indicates that small models can reliably replace large LLM-based compressors when equipped with optimized guidelines. These results confirm that small models are sufficient for compression, enabling the expensive LLM to be only reserved for the agent.

### 4.4. ACON for distilled small agents

We examine whether ACON also benefits smaller LLM agents, which are particularly vulnerable to long-horizon inefficiency. Without compression, models such as Qwen3-14B often fail on medium and hard tasks due to distracting context. As shown in Figure 5, ACON substantially improves their performance: on AppWorld, Qwen3-14B achieves a **32.4% relative improvement** (from 25.6% to 33.9%), and on 8-objective QA, it shows a **45.6% gain** (from 0.158 to 0.23 EM). These results demonstrate that ACON acts as an equalizer, enabling smaller agents with concise but informative contexts to approach the performance of larger models.

### 4.5. Analysis

**Compression threshold: moderate value yields the best trade-off.** In Figure 6, we provide ablations on threshold for compression in Equation 3 and Equation 4. Results show that smaller thresholds reduce tokens but incur more frequent compression calls and degrade accuracy, while larger thresholds preserve accuracy with higher cost. Moderate values (4096 for history, 1024 for observation) provide the best trade-off, maintaining accuracy close to no compression while still reducing peak tokens substantially.

*Table 3.* **Ablation studies on the prompt optimizer** in App-World, gpt-4.1 agent and history compressor. Default is o3 optimizer with task contrastive feedback.

| Optimizer model | Task contrastive | Average Acc. |
|---|---|---|
| **o3** | ✓ | **51.2** |
| o3 | ✗ | 50.6 (-0.6) |
| gpt-4.1 | ✓ | 47.6 (-3.6) |
| gpt-5 | ✓ | 50.6 (-0.6) |

**Prompt optimizer: o3 + contrastive feedback works best.** We analyze how the choice of optimizer and the use of contrastive feedback affect compression guideline quality. As shown in Table 3, the default o3 with contrastive feedback yields the best performance, while removing contrastive feedback (only using failed trajectories) or switching to other models results in lower accuracy. Although o3 shows the best performance, we also demonstrate that the optimizer model can be replaced to weaker models such as gpt-4.1, showing it still yields sufficiently fine guideline compared to the baseline guideline.

**Optimization cost: practical and lightweight.** One might assume that leveraging a reasoning model like o3 would be cost-prohibitive. However, our guideline optimization step is remarkably lightweight, demanding *less than $2 per benchmark*. This can be negligible compared to the expense of trajectory rollouts on the training dataset. For instance, utilizing gpt-4.1 on AppWorld train set requires a total rollout cost of approximately $20. Furthermore, even this data collection cost is substantially lower than that of RL-based methods such as GRPO (Shao et al., 2024), which requires extensive rollout for advantage estimation. Detailed API cost computation and breakdowns are provided in Section B.2 and C.2.

**API cost analysis of compressors across different models.** While using a compressor reduces context length, it incurs additional computational costs beyond the agent. In Section 3.4, we proposed a distillation strategy to mitigate this overhead. To quantify the cost efficiency, we analyze the expenses using API pricing as a proxy. With gpt-4.1 as the baseline compressor, the cost remains significant at $0.045 per example. Switching to gpt-4.1-mini reduces the compression cost to $0.014, achieving a 69.2% reduction. However, the most substantial gain is observed with distilled Qwen3-14B, where the cost decreases to $0.0004. This represents a 99.1% reduction compared to the teacher model, effectively minimizing the cost burden of context compression.

**Practical efficiency trade-off.** We further quantify the practical overhead of generative compression on App-World. For API cost, we estimate the end-to-end cost of

*Table 4.* **Practical efficiency on AppWorld.** API cost is estimated per task with a gpt-4.1 agent and Qwen3-14B compressor. Latency is measured as median wall-clock time per task with a Qwen3-14B agent and Qwen3-14B compressor.

| Method | API Cost ↓ | Latency ↓ |
|---|---|---|
| No Compression | $0.331 | 73.24s |
| ACON (history) | $0.285 | 87.68s |
| ACON (observation) | $0.272 | 101.92s |

both the agent and compressor, excluding input caching. For latency, we run the agent and compressor locally on a single A100 GPU rather than using API endpoints, which avoids confounding factors such as network overhead and server-side queueing. As shown in Table 4, ACON reduces end-to-end API cost by compressing the context of agent, but introduces additional wall-clock latency due to the extra compressor call.

We include more experimental results including experiments with different agent models, additional ablation studies, case study, and qualitative examples of context compression in Appendix C and D.

## 5. Conclusion

We presented **Agent Context Optimization (ACON)**, a unified framework that systematically compresses both interaction histories and environment observations for long-horizon LLM agents. Unlike prior work that relies on naive prompting or narrow domains, ACON introduces compression guideline optimization in natural language space, enabling adaptive and model-agnostic compression. Experiments on AppWorld, OfficeBench, and Multi-objective QA show that ACON reduces peak tokens by 26–54% while improving task success over existing compression baselines, with small degradation relative to full-context baseline. Beyond memory efficiency, we demonstrate that optimized compressors can be distilled into smaller models, substantially lowering overhead without sacrificing performance. Moreover, by supplying concise yet informative contexts, ACON allows small agents such as Qwen3-14B to approach the performance of much larger models. Overall, our findings highlight that ACON lays a foundation for more general, memory-efficient, and deployable long-horizon LLM agents.

**Limitations and Future Work.** While ACON effectively reduces context costs, a few limitations remain. First, our empirical evaluation primarily focuses on GPT models due to resource constraints. Second, like many context management frameworks, the compression process itself introduces computational overhead and increased latency. We include more detailed discussion on limitations and future work in Appendix A.

## Impact Statement

This paper presents work whose goal is to advance the field of Large Language Models, specifically by addressing the computational costs of long-horizon autonomous agents. By optimizing context compression and enabling effective distillation into smaller models, our work primarily contributes to making advanced agentic capabilities more resource-efficient and accessible. This has positive implications for reducing the computational footprint of AI systems and lowering barriers to entry for researchers with limited resources. We do not foresee specific negative societal consequences beyond the general considerations required when deploying autonomous systems.

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

## A. Limitations & Future Works

Our work addresses the context management problem in long-horizon LLM agents and proposes a framework for optimizing context compression. While the method effectively reduces token costs with minimal performance degradation, it also presents several limitations and potentials for future works.

**Scope of empirical evaluation.**   Our work primarily focuses on the GPT models due to computational and budgetary constraints. While the proposed framework is designed to be model-agnostic, its empirical generalizability to other foundation models, such as Gemini (Comanici et al., 2025) or Claude (Anthropic, 2024), has yet to be extensively verified. Furthermore, the inclusion of massive open-source models like DeepSeek-R1 (DeepSeek-AI et al., 2025) or Qwen3-235B (Yang et al., 2025) was restricted by GPU availability. Extending the analysis to these models would provide stronger evidence of robustness and broaden the applicability of our conclusions.

**Real-world deployability**   Although our validation on three distinct benchmarks reflects realistic agentic scenarios, these environments remain controlled. Transitioning from benchmark-centric evaluations to in-the-wild deployment—where task complexity is stochastic and environmental constraints are less predictable—remains an open challenge. We identify the integration of our framework into live, multi-agent production systems as a valuable next step.

**Convergence of natural language space optimization**   The optimization of our compression guidelines relies on LLM-generated feedback, a strategy aligned with recent prompt optimization work (Yüksekgönül et al., 2025; Khattab et al., 2024; Pryzant et al., 2023; Yang et al., 2024a). However, unlike traditional numerical gradient-based optimization, this approach lacks a formal convergence guarantee. While our method of sampling multiple candidates and selecting the best performer partially mitigates this, it does not provide a principled theoretical foundation. A deeper analysis of the natural language space optimization process and more principled methods for optimizing the objective in 3.3 would be valuable directions for future work.

**Distillation gap and data scalability**   A performance gap remains between our distilled models and their teacher models. We hypothesize that our current dataset (100 examples per domain) may not fully capture the reasoning required for complete behavioral cloning. Future work will investigate whether increasing the diversity and volume of training data can eliminate the residual performance gap.

**Latency and KV-cache dynamics**   A significant, yet often overlooked, challenge is the computational overhead inherent in the compression process itself—a limitation shared by the majority of existing context compression frameworks (Lee et al., 2025; Smith, 2025; Yang et al., 2024b; Sun et al., 2025; Zhou et al., 2025; Ye et al., 2025). In transformer-based architectures, history compression typically invalidates the existing KV-cache, necessitating a costly re-computation of the entire compressed sequence. While observation compression can mitigate this by reducing the context before caching them, the generative overhead required for the compression procedure itself still introduces latency, often slowing down the agent's response time (Lee et al., 2025). Consequently, the field currently faces a trade-off where the gains in long-horizon efficiency are partially offset by the increased computational cost of the compression itself. To move beyond these constraints, the development of KV-cache eviction or compression strategies—specifically optimized for the dynamic, multi-turn nature of agentic workflows—can be a potential avenue for future research, building upon foundational works in long-context modeling (Zhang et al., 2025; Willette et al., 2025; Kim et al., 2025).

## B. Experimental Setup Details

### B.1. Datasets

**AppWorld (Trivedi et al., 2024).**   AppWorld is our primary benchmark, providing a high-quality execution environment that integrates nine everyday applications (Spotify, SimpleNote, Amazon, Venmo, Gmail, Splitwise, File system, and Phone) through 457 APIs. It also includes realistic simulations of approximately 100 functional users. This benchmark is particularly suitable for evaluating long-horizon productivity agents, as its multi-step tasks require an average of 42.5 API calls per task. We follow the official split, using 90 training tasks for guideline optimization and distillation, and 168 test-normal tasks for evaluation. An example trajectory from AppWorld is provided in Example D.1.

*Table 5.* Example tasks across benchmarks.

| Benchmark / Difficulty | Example Task |
| --- | --- |
| **AppWorld** | |
| Easy | Mark "Taking a solo backpacking trip" in my Bucket List Simple Note as not done. |
| Medium | Like all the Venmo transactions from today involving any of my roommates on my Venmo social feed. |
| Hard | Start playing a playlist on Spotify that has enough songs for my workout today. I do not want to have to change the playlist in the middle of my workout. The workout plan is in Simple Note. |
| **OfficeBench** | |
| 1-app | Create a new Word file called `random_paragraph.docx` and add the content in `random_paragraph.txt` to it. |
| 2-app | Analyze Excel data of students' grade and generate a teaching report in `teaching.docx`. |
| 3-app | Read company revenues and send an email with subject `revenues`, containing data to Bob for reporting, also write a `revenue.docx` to summarize it. |
| **8-objective QA** | |
| – | who wrote the song *Oceans Where Feet May Fail*?; who plays Eddie the Eagle in the movie?; when was the last time England were in the final of World Cup?; who plays Chelsea's mom on *Young and the Restless*?; what is the largest coin in the US?; who sang *Even the Bad Times Are Good*?; who sings *This Is My Town* country song?; which of the Guianas is not an independent country? |

**OfficeBench (Wang et al., 2024b).** OfficeBench is a benchmark for office automation using applications such as Word, Excel, PDF, Calendar, Email, Shell, and Calculator. It evaluates the ability of agents to coordinate across multiple apps to complete complex tasks, making it well suited for long-horizon scenarios. Tasks are categorized as 1-app, 2-app, or 3-app depending on the number of applications required. We restrict our experiments to text-related tasks, excluding those requiring OCR, as OCR quality could confound the evaluation. Since no official split is available, we randomly partition the tasks into training and test sets with a 1:1 ratio, resulting in 92 training tasks and 95 test tasks. We additionally refine the dataset by removing ambiguous tasks and ensuring that synthetic files (testbeds) are not shared across splits.

**8-Objective QA (Zhou et al., 2025).** The 8-objective QA benchmark simulates deep research-style agentic tasks. Unlike conventional multi-hop QA, which requires answering a single question using multiple pieces of evidence, this benchmark poses eight distinct questions within one task, and the agent must provide answers to all of them at the end. This design creates a more challenging setting for long-horizon agents. Following Zhou et al. (2025), we construct each task by grouping eight questions together. Questions are drawn from NaturalQuestions (Kwiatkowski et al., 2019), resulting in 100 training tasks (from the train split) and 100 test tasks (from the test split). For retrieval, we use a BM25 retriever over the 2018 Wikipedia knowledge base, following Jin et al. (2025).

We include the example task of each benchmark in Table 5.

### B.2. Evaluation Metrics

For efficiency evaluation, we adopt two metrics—*peak tokens* and *dependency*—introduced in LightThinker (Zhang et al., 2025) and MEM1 (Zhou et al., 2025).

**Peak tokens.** Peak tokens are measured as the maximum number of tokens observed in any single sequence throughout the agent's trajectory, excluding system prompts. This metric serves as a proxy for inference-time memory requirements and corresponds to the maximum peak shown in Figure 2.

**Dependency.** Dependency is defined as the area under the curve in Figure 2. At each step $t$, given the number of input tokens $n_i^{(t)}$ and output tokens $n_o^{(t)}$, it is calculated as:

$$\sum_{t \in [T]} \frac{(n_i^{(t)} + 2n_o^{(t)}) \times n_o^{(t)}}{2}. \tag{12}$$

This metric approximates the cumulative computational cost incurred by action generation across the trajectory.

**API Cost.** For the cost analysis, we use the official OpenAI pricing (as of September 2025) for `gpt-4.1` and `gpt-4.1-mini` (OpenAI, 2025b). Specifically, `gpt-4.1` is priced at \$3.00 per 1M input tokens and \$12.00 per 1M output tokens. For `gpt-4.1-mini`, the costs are \$0.80 per 1M input tokens and \$3.20 per 1M output tokens. For `Qwen3-14B` (Yang et al., 2025), since no official API pricing is available, we approximate the cost using OpenRouter[1] : \$0.06 per 1M input tokens and \$0.24 per 1M output tokens.

### B.3. Implementation Details & Hyperparameters

**API Inference.** We set temperature 0.0 and fix the seed 42. Note that there is still non-determinism with fixing the seed and setting temperature as 0. To reduce the instability, we use the API snapshot form Azure OpenAI endpoint `gpt-4.1-2025-04-14` and `gpt-4.1-mini-2025-04-14`.

**Compression.** For history compression, we set $T_{\text{hist}} = 4096$ for AppWorld and OfficeBench, and 2048 for 8-objective QA. We keep the last action, observation pair to preserve the latest information. This is the same for ACON and all baselines. For observation compression, we set $T_{\text{obs}} = 1024$ for AppWorld, 512 for OfficeBench, and 400 for 8-objective QA.

**Prompt Optimization.** We use the OpenAI `o3` model (OpenAI, 2025d) for both analysis and update of prompts. During the update stage, we sample 5 candidate prompts and select the one that performs best on a subset of the training set.

The prompts used in each stage and step are provided as follows:

- Analysis prompt for UT step: Prompt D.1
- Update prompt for UT step: Prompt D.2
- Analysis prompt for CO step: Prompt D.3
- Update prompt for CO step: Prompt D.4

We also provide the detailed procedure in Algorithm 1. For the subset used in prompt selection during the UT step, we consider training tasks in $\mathcal{D}_{\text{cont}}^{(r)}$ where the agent succeeds without compression but fails with compression. For the CO step, we use training tasks in $\mathcal{D}_{\text{succ}}^{(r)}$ where the agent succeeds with compression. We perform one round consisting of a single UT step and a single CO step to obtain the guidelines used in our experiments, unless otherwise noted.

**Baselines** For FIFO, we keep last 5 interaction turns which fits in similar compression rate in average with ACON. For retrieval, we also retrieve 4 interaction turns and keep the last turn. We use OpenAI `text-embedding-3-large` for embedding. For LLMLingua, we set keep rate as 30% for both history and observation. For naive prompting, we use the similar prompt with Lee et al. (2025) and do some human prompt engineering to specialize each prompt to history or observation compression.

**Compressor & Agent Distillation** Both compressor and agent distillation use LoRA (Hu et al., 2022) with rank 16, $\alpha = 32$, learning rate $10^{-4}$, 3 epochs, batch size 4, and maximum sequence length 10,000. We adopt linear warmup (5% ratio), weight decay 0.01, and AdamW optimizer. No hyperparameter tuning was performed; the same setup is applied across all models and benchmarks. We sample a single generation from the teacher for fine-tuning, leaving potential improvements from hyperparameter tuning or multi-sample training for future work. We use 1 A100 80GB GPU for both training and inference. For inference of fine-tuned models, we use greedy decoding (temperature 0.0).

---

[1] https://openrouter.ai/

## C. Additional Results

We provide additional quantitative results to complement the main experiments in Section 4.

### C.1. Results with different agent models

**Experiments with gpt-4.1-mini.** Results for the smaller variant gpt-4.1-mini (OpenAI, 2025b) across three benchmarks are reported in AppWorld (Table 9), OfficeBench (Table 10), and 8-objective QA (Table 11). The trends of ACON are consistent with those for gpt-4.1 in Section 4. In particular, Table 9 shows that history compression improves the performance of gpt-4.1-mini compared to the baseline, complementing the findings in Section 4.4 that ACON enhances the effectiveness of smaller LM agents. These results highlight the robustness of our method under resource-constrained settings.

**Experiments with gpt-5-chat.** We also evaluate on AppWorld using gpt-5-chat (OpenAI, 2025c), as reported in Table 12. The improvements follow the same trend as with gpt-4.1, demonstrating that ACON generalizes to the latest stronger proprietary models.

### C.2. Detailed results and analyses

**OfficeBench difficulty breakdowns.** We further analyze OfficeBench with gpt-4.1 by difficulty level. The detailed breakdown in Table 8 shows that ACON yields the largest gains on the most challenging tasks in Level 3.

**Distilled optimizer.** Additional results for the distilled optimizer in AppWorld are shown in Table 13. Beyond the analysis in Section 4.3, we also include experiments where the compressor is distilled using guidelines without optimization. The results confirm that optimized guidelines consistently yield stronger performance when distilled into smaller models.

**History and observation compression.** In Table 14, we report ablations with gpt-4.1 using both history and observation compression. While combining the two compressions achieves larger reductions in peak token usage and dependency, it also leads to substantial performance degradation compared to applying either compression alone.

**Additional guideline optimization step.** We investigate whether running an extra utility maximization step ($\overline{\text{UT}}$) after the standard sequence of utility maximization and compression maximization ($\overline{\text{CO}}$) is beneficial. As shown in Table 14, this additional iteration results in a performance drop, indicating that a single round of optimization is sufficient for effective guideline learning.

**Distilled compressor for observation.** In addition to Section 4.3, we report results for observation compressor distillation in Figure 7. Similar to history compression, the performance is largely preserved after distillation, confirming that ACON enables effective transfer of optimized observation compressors to smaller models.

**API cost details on compression guideline optimization.** In this section, we provide a detailed breakdown of the computational costs associated with our framework. All cost estimates are based on the official API pricing as in Section B.2. The total expense is categorized into two phases: trajectory rollout and guideline optimization. *(1) Trajectory Rollout (Data Collection).* This phase accounts for the majority of the budget. We utilize `gpt-4.1` to collect trajectories on the training set (*e.g.,* 90 examples for AppWorld). For each example, we generate two trajectories: one without compression and one with compressed context. The total cost for collecting these rollout trajectories amounts to approximately $20. It is important to note that this is a one-time data preparation cost which can be amortized across future agent runs or substituted with existing offline logs. *(2) Guideline Optimization.* Despite utilizing the reasoning-intensive o3 model, the optimization process is highly cost-efficient. The procedure runs for a single number of iteration. In each iteration, the optimizer generates 5 candidate guidelines and performs one $\overline{\text{UT}}$ step and one $\overline{\text{CO}}$ step. The total cost for the entire optimization loop is consistently under $2 per domain.

**Additional Results on WebVoyager** To further evaluate ACON beyond simulated productivity and question-answering environments, we conduct additional experiments on WebVoyager (He et al., 2024), a web-agent benchmark that requires agents to interact with webpages and complete user-specified tasks. We use 50 training tasks for compression guideline optimization and evaluate the resulting guidelines on 70 held-out test tasks. We evaluate WebVoyager using gpt-4.1 as both the agent and compressor, following the main experimental setting.

*Table 7.* Comparison between ACON and MEM1-like methods. The two methods operate under different assumptions regarding model accessibility, training requirements, and architectural coupling.

| Dimension | ACON (ours) | MEM1-like (Zhou et al., 2025; Sun et al., 2025; Ye et al., 2025) |
|---|---|---|
| Is agent model training not required? | ✓ no agent model training or weight updates required | ✗ requires RL training on the agent model |
| Can the method work without access to model weights? | ✓ works with both open-source and proprietary API models | ✗ requires full model access and gradients |
| Can the agent and compressor be different models? | ✓ supports decoupled design with different model sizes | ✗ reasoning and compression are integrated into a single model |
| Is it possible to use a large agent with a small compressor? | ✓ supports combinations like gpt-4.1 agent + Qwen3-14B compressor | ✗ same model must serve as both agent and compressor |
| Does optimization avoid GPU-based RL cost? | ✓ under $2 for guideline optimization, no GPU needed | ✗ RL policy training requires multiple trajectories and GPU computation |

This setting introduces different context dynamics from AppWorld, OfficeBench, and 8-objective QA, as the agent must process verbose webpage observations represented as accessibility trees (AXTrees) while maintaining relevant interaction history across multiple steps (Lee et al., 2025).

As reported in Table 6, ACON improves task accuracy over both no compression and prompting baselines while substantially reducing peak tokens and dependency compared to no compression. For history compression, ACON improves accuracy from 42.9 to 48.6 compared to prompting, with comparable peak tokens and lower dependency. For observation compression, ACON improves accuracy from 45.7 to 47.1 and reduces dependency from 0.85 to 0.80. These results suggest that the optimized compression guidelines learned by ACON are effective not only in structured tool-use environments, but also in web-agent settings where observations are verbose and dynamically changing.

*Table 6.* Additional results on WebVoyager. We report task accuracy, peak input tokens ($10^3$), and dependency ($10^6$).

| Method | Acc. ↑ | Peak ↓ | Dep. ↓ |
|---|---|---|---|
| No compression | 35.7 | 13.28 | 2.53 |
| **History Compression** | | | |
| Prompting | 42.9 | **7.97** | 1.23 |
| ACON | **48.6** | 8.04 | **1.19** |
| **Observation Compression** | | | |
| Prompting | 45.7 | **3.91** | 0.85 |
| ACON | **47.1** | 4.19 | **0.80** |

**Case study: history compression turns failure into success.** A notable case study illustrates how history compression enables a smaller agent to succeed on tasks that would otherwise fail. In the uncompressed trajectory in Example D.2, the gpt-4.1-mini agent repeatedly attempted to use the `file_system` APIs without managing authentication, leading to persistent 401 Unauthorized errors. After compressing the history as in Example D.3, however, the compressed history retained only the essential reasoning steps: the need for both username and password, the importance of passing the returned `access_token` into subsequent calls, and the absence of proxy APIs in the supervisor app.

This compressed context prevented redundant exploration and guided the agent directly to the correct sequence—login with full credentials, capture the token, and provide it explicitly in `show_directory` and `delete_file` calls. As a result, the agent was able to enumerate and remove all .pdf files in /downloads, a task it had previously failed. This example highlights how compression does not merely shorten history but clarifies critical dependencies, turning a failure trajectory into a successful one.

## C.3. Comparison with MEM1

MEM1 (Zhou et al., 2025) and concurrent works (Sun et al., 2025; Ye et al., 2025) propose a learnable context compression policy trained jointly with the agent through reinforcement learning. This design couples reasoning and compression within a single trainable model and requires full access to model weights and gradient updates. In contrast, our method can perform optimization entirely at the prompt-level without any weight updates and enables the agent and compressor to be different models.

This decoupling allows combinations that are not possible in MEM1 and other similar methods. For example, one can use a large proprietary model such as gpt-4.1 as the agent while employing a smaller open-source model such as Qwen3-14B as the compressor after distillation, a configuration that other MEM1-like methods cannot support due to its unified training

requirement. This flexibility makes ACON applicable to both open-source and proprietary API-based models, including settings where model weights are inaccessible. A detailed comparison is summarized in Table 7.

## D. Qualitative Examples

We complement the quantitative results with qualitative illustrations.

**Compression guidelines.**   We present examples of compression guidelines before and after optimization in AppWorld. The history compression guideline before optimization is shown in Prompt D.5, the optimized version (UT) in Prompt D.6, and the optimized version (UTCO) in Prompt D.7. Similarly, observation compression guideline examples are provided in Prompt D.8 and Prompt D.9, and the optimized version (UTCO) in Prompt D.10. These comparisons demonstrate that optimization yields more targeted guidelines for compressors.

**Compressed histories.**   Compression Example D.1 illustrates history segments before and after guideline optimization in AppWorld with gpt-4.1. The optimized guideline retains a more detailed record of task progress, including variable states and guardrails for the environment. After the compression maximization step (CO), the histories become shorter while still preserving the essential information required for future decision-making. This qualitative evidence demonstrates how our framework improves both the efficiency and effectiveness of context compression, complementing the guideline optimization procedure described in Section 3.3.

We also present Compression Example D.2 for 8-objective QA and Compression Example D.3 for OfficeBench, which confirm that the effects of guideline optimization are consistent across benchmarks.

**Compressed observations.**   Compression Example D.4 shows observations before and after guideline optimization in AppWorld. We illustrate the case of printing available APIs for the Spotify app, which produces a lengthy observation. The optimized guideline yields a more structured and faithful representation: whereas naive prompting loses the JSON format and omits the crucial "play_music" API, the optimized version preserves both structure and key functionality necessary to complete the task.

*Table 8.* Detailed results on **OfficeBench** benchmark. We report accuracy (%), and efficiency metrics: average steps, peak input tokens ($10^3$), and dependency ($10^6$) for Average and each difficulty level. Best values are in bold. Rows in blue background indicate the results from **ours** .

| Method | Average (All) | | | | Level 1 (1-app, 42) | | | Level 2 (2-app, 22) | | | Level 3 (3-app, 31) | | |
|---|---|---|---|---|---|---|---|---|---|---|---|---|---|
| | Acc. ↑ | Steps ↓ | Peak ↓ | Dep. ↓ | Acc. ↑ | Peak ↓ | Dep. ↓ | Acc. ↑ | Peak ↓ | Dep. ↓ | Acc. ↑ | Peak ↓ | Dep. ↓ |
| **Agent: gpt-4.1 / Compressor: gpt-4.1** | | | | | | | | | | | | | |
| No Compression | **76.84** | 11.52 | 7.27 | 4.43 | **92.86** | 6.23 | 4.05 | 77.27 | 6.14 | 1.81 | 54.84 | 8.37 | 6.08 |
| **History Compression** | | | | | | | | | | | | | |
| FIFO | 67.37 | 12.26 | **4.02** | 2.64 | 83.33 | 4.19 | 0.72 | 63.64 | **3.51** | **1.01** | 48.39 | **4.23** | 4.39 |
| Retrieval | 65.26 | 16.20 | 4.33 | 2.06 | 85.71 | 4.35 | 0.84 | 63.64 | 3.52 | 1.37 | 38.71 | 4.78 | 2.99 |
| LLMLingua | 70.53 | 10.89 | 4.65 | 1.85 | 83.33 | 4.17 | **0.67** | 68.18 | 4.61 | 1.18 | 54.84 | 4.88 | 2.74 |
| Prompting | 71.58 | **10.13** | 4.40 | **1.10** | 85.71 | 4.18 | 0.81 | 77.27 | 4.53 | 1.08 | 48.39 | 4.42 | **1.23** |
| ACON UT | 74.74 | 13.13 | 4.93 | 3.85 | 85.71 | 4.71 | 6.89 | 72.73 | 4.64 | 1.44 | 61.29 | 5.19 | 3.89 |
| ACON UTCO | 72.63 | 11.54 | 4.54 | 1.91 | 88.10 | 3.92 | 0.76 | 72.73 | 4.72 | 1.16 | 51.61 | 4.71 | 2.84 |
| **Observation Compression** | | | | | | | | | | | | | |
| LLMLingua | 71.58 | 11.89 | 7.38 | 6.14 | 80.95 | 7.35 | 12.40 | 72.73 | 6.31 | 2.11 | **58.06** | 7.99 | 5.70 |
| Prompting | 55.79 | 12.24 | 6.44 | 2.68 | 78.57 | 4.51 | 0.98 | 50.00 | 6.98 | 2.61 | 29.03 | 6.98 | 3.46 |
| ACON UT | 73.68 | 10.83 | 6.55 | 3.85 | 90.48 | 6.57 | 8.02 | 77.27 | 6.11 | 1.97 | 48.39 | 6.80 | 3.10 |
| ACON UTCO | 72.63 | 10.28 | 6.17 | 2.88 | 88.10 | 4.75 | 0.82 | 72.73 | 6.41 | 2.09 | 51.61 | 6.65 | 4.22 |

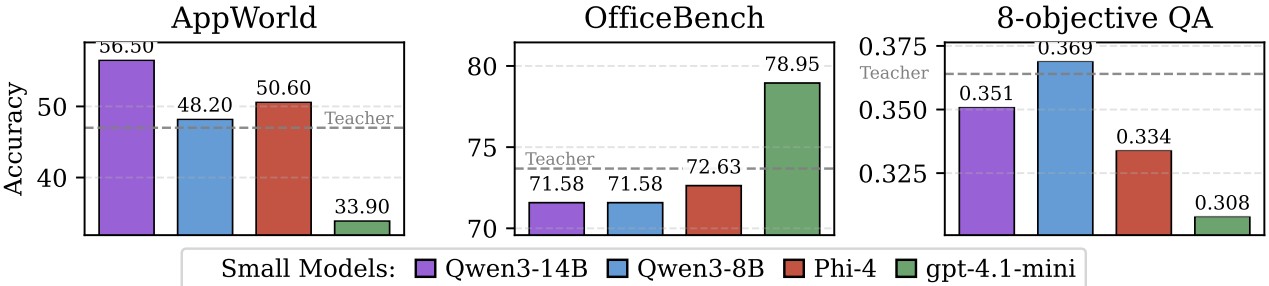

*Figure 7.* **Results of distilled compressors on observation compression** with gpt-4.1 as the agent. Student models (Qwen3-14B, Qwen3-8B, Phi-4) are distilled from gpt-4.1 compressor using the optimized compression guideline after UT step, and evaluated across all benchmarks. We also include result with gpt-4.1-mini without distillation for comparison.

*Table 9.* Results across different difficulty levels on **Appworld** benchmark (test-normal) with gpt-4.1-mini. We adopt the same compression guidelines as those used in the gpt-4.1 experiments. Each block reports accuracy (task goal completion score), average steps, average peak input tokens ($10^3$), and average dependency ($10^6$) for agents. Best results in each column are highlighted in bold. Rows in blue background indicate the results from **ours** .

| Method | Average (168) | | | | Easy (57) | | | Medium (48) | | | Hard (63) | | |
|---|---|---|---|---|---|---|---|---|---|---|---|---|---|
| | Acc. ↑ | Steps ↓ | Peak ↓ | Dep. ↓ | Acc. ↑ | Peak ↓ | Dep. ↓ | Acc. ↑ | Peak ↓ | Dep. ↓ | Acc. ↑ | Peak ↓ | Dep. ↓ |
| **Agent: gpt-4.1-mini / Compressor: gpt-4.1-mini** | | | | | | | | | | | | | |
| No compression | 35.7 | 18.14 | 8.55 | 5.07 | 56.1 | 6.45 | 3.72 | 31.2 | 8.31 | 4.79 | 20.6 | 10.64 | 9.18 |
| **History Compression** | | | | | | | | | | | | | |
| FIFO | 39.3 | 30.39 | 6.18 | 5.24 | 75.4 | 4.76 | 2.66 | 35.4 | 5.33 | 4.81 | 9.5 | 8.10 | 7.91 |
| Retrieval | 14.9 | 40.18 | 7.49 | 5.95 | 36.8 | 7.10 | 4.29 | 8.3 | 7.44 | 6.80 | 0.0 | 7.89 | 6.81 |
| LLMLingua | 36.3 | 28.41 | 7.24 | 6.65 | 66.7 | 6.96 | 3.84 | 33.3 | 7.05 | 7.60 | 11.1 | 7.62 | 8.47 |
| Prompting | 35.7 | 24.98 | 6.56 | 4.95 | 64.9 | 5.96 | 2.90 | 27.1 | 6.65 | 5.35 | 15.9 | 6.84 | 6.49 |
| ACON UT | 42.3 | 22.46 | 6.51 | 5.48 | 64.9 | 5.87 | 2.62 | 37.5 | 7.18 | 5.22 | 25.4 | 7.18 | 8.25 |
| ACON UTCO | 32.7 | 24.27 | 6.99 | 4.97 | 57.9 | 7.50 | 2.77 | 33.3 | 8.45 | 4.99 | 9.5 | 6.95 | 6.97 |
| **Observation Compression** | | | | | | | | | | | | | |
| LLMLingua | 25.6 | 20.75 | 8.04 | 8.21 | 38.6 | 6.13 | 3.03 | 27.1 | 8.74 | 13.78 | 12.7 | 9.24 | 8.65 |
| Prompting | 33.9 | 16.71 | 6.04 | 3.87 | 59.7 | 5.21 | 3.41 | 33.3 | 5.99 | 3.27 | 11.1 | 6.83 | 4.74 |
| ACON UT | 33.9 | 16.78 | 6.86 | 4.58 | 59.7 | 5.44 | 2.93 | 33.3 | 7.13 | 4.26 | 11.1 | 7.97 | 6.38 |
| ACON UTCO | 27.4 | 17.89 | 6.37 | 4.44 | 40.4 | 5.18 | 2.40 | 35.4 | 6.84 | 5.03 | 9.5 | 7.09 | 5.82 |

---

**Algorithm 1** Compression Guideline Optimization ($\underline{\text{UT}} \leftrightarrow \underline{\text{CO}}$)

---

1: **Input:** training indices $\mathcal{I}$; fixed agent $\mathcal{M}(\cdot; \theta, \mathcal{P}_{\text{agent}})$; compressor $f(\cdot; \phi, \mathcal{P})$; initial guideline $\mathcal{P}^{(0)}$; tradeoff $\lambda \geq 0$;
   rounds $R$; candidates $K$
2: **Output:** optimized guideline $\mathcal{P}^*$
3: **Notation:** for each $i \in \mathcal{I}$ and guideline $\mathcal{P}$:
4:     baseline (no compression): context $\boldsymbol{H}_i$ with success $r_i^{\text{base}} \in \{0, 1\}$
5:     compressed: $\boldsymbol{H}_i'(\mathcal{P})$ with success $r_i(\mathcal{P}) \in \{0, 1\}$ and cost
6:     $C(\boldsymbol{H}_i'(\mathcal{P})) = \sum_t \mathcal{C}(\boldsymbol{h}_{i,t-1}', o_{i,t}')$
7: **# 0) Collect baseline contexts (no compression)**
8: **for all** $i \in \mathcal{I}$ **do**
9:     Run $\mathcal{M}$ without compression to obtain $\boldsymbol{H}_i$ and $r_i^{\text{base}}$
10: **end for**
11: $\mathcal{I}^+ \leftarrow \{i \in \mathcal{I} \mid r_i^{\text{base}} = 1\}$                              *# indices where baseline succeeds*
12: **for** $r = 0$ **to** $2R - 2$ **do**
13:     **# Note:** $r$ takes even values only (i.e., $0, 2, \ldots, 2R - 2$)
14:     **# Stage A: $\underline{\text{UT}}$ (reward-first update using $H$ vs. $H'$)**
15:     **for all** $i \in \mathcal{I}$ **do**
16:         Run $\mathcal{M}$ with compression $f(\cdot; \phi, \mathcal{P}^{(r)})$ to obtain $\boldsymbol{H}_i'(\mathcal{P}^{(r)})$, $r_i(\mathcal{P}^{(r)})$, $C(\boldsymbol{H}_i'(\mathcal{P}^{(r)}))$
17:     **end for**
18:     $\mathcal{D}_{\text{cont}}^{(r)} \leftarrow \{(\boldsymbol{H}_i, \boldsymbol{H}_i'(\mathcal{P}^{(r)})) \mid i \in \mathcal{I}^+, \ r_i(\mathcal{P}^{(r)}) = 0\}$
19:     **for all** $(\boldsymbol{H}, \boldsymbol{H}') \in \mathcal{D}_{\text{cont}}^{(r)}$ **do**
20:         *# contrastive feedback: what did $H'$ miss vs. $H$?*
21:         Feedback $\leftarrow$ LLM(FeedbackInstr, $\boldsymbol{H}, \boldsymbol{H}'$)
22:         Append Feedback to multiset $\mathcal{F}_{\text{util}}$
23:     **end for**
24:     $\{\mathcal{P}_k^{(r+1)}\}_{k=1}^K \leftarrow$ LLM(UpdateInstr, $\mathcal{P}^{(r)}, \|_{f \in \mathcal{F}_{\text{util}}} f$)
25:     *# where $\|$ denotes concatenation*
26:     **# Select by reward:** evaluate on a held-out subset of $\mathcal{I}^+$ and pick
27:     $k_{\text{util}}^* \leftarrow \arg\max_k \ \text{SuccessRate}(\{r_i(\mathcal{P}_k^{(r+1)})\}_{i \in \mathcal{I}^+})$
28:     $\mathcal{P}_{\text{util}}^{(r+1)} \leftarrow \mathcal{P}_{k_{\text{util}}^*}^{(r+1)}$
29:     **# Stage B: $\underline{\text{CO}}$ (cost-minimizing refinement using only $H'$)**
30:     **for all** $i \in \mathcal{I}$ **do**
31:         Using $\mathcal{P}_{\text{util}}^{(r+1)}$, obtain $\boldsymbol{H}_i', r_i, C(\boldsymbol{H}_i')$
32:     **end for**
33:     $\mathcal{D}_{\text{succ}}^{(r)} \leftarrow \{\boldsymbol{H}_i' \mid r_i = 1\}$
34:     **for all** $\boldsymbol{H}' \in \mathcal{D}_{\text{succ}}^{(r)}$ **do**
35:         *# find redundant spans within $H'$*
36:         CompFeedback $\leftarrow$ LLM(CompressInstr, $\boldsymbol{H}'$)
37:         Append CompFeedback to multiset $\mathcal{F}_{\text{comp}}$
38:     **end for**
39:     $\{\tilde{\mathcal{P}}_k^{(r+2)}\}_{k=1}^K \leftarrow$ LLM(UpdateInstr, $\mathcal{P}_{\text{util}}^{(r+1)}, \|_{f \in \mathcal{F}_{\text{comp}}} f$)
40:     **# Select by reward-cost:** evaluate on a held-out split of $\mathcal{I}$ and pick
41:     $k_{\text{comp}}^* \leftarrow \arg\max_k \left( \text{SuccessRate}(\{r_i(\tilde{\mathcal{P}}_k^{(r+2)})\}) - \lambda \cdot \text{NormCost}(\{C(\boldsymbol{H}_i'(\tilde{\mathcal{P}}_k^{(r+2)}))\}) \right)$
42:     $\mathcal{P}^{(r+2)} \leftarrow \tilde{\mathcal{P}}_{k_{\text{comp}}^*}^{(r+2)}$
43:     **if** early-stop criterion satisfied **then**
44:         **break**                               *# e.g., success/cost convergence or budget met*
45:     **end if**
46: **end for**
47: $\mathcal{P}^* \leftarrow \mathcal{P}^{(r+2)}$
48: **return** $\mathcal{P}^*$

---

*Table 10.* Detailed results on **OfficeBench** benchmark with gpt-4.1-mini. We adopt the same compression guidelines as those used in the gpt-4.1 experiments. We report accuracy (%), and efficiency metrics: average steps, peak input tokens ($10^3$), and dependency ($10^6$) for Average and each difficulty level. Rows in blue background indicate the results from `ours`.

| Method | Average (All) | | | | Level 1 (1-app, 42) | | | Level 2 (2-app, 22) | | | Level 3 (3-app, 31) | | |
|---|---|---|---|---|---|---|---|---|---|---|---|---|---|
| | Acc. ↑ | Steps ↓ | Peak ↓ | Dep. ↓ | Acc. ↑ | Peak ↓ | Dep. ↓ | Acc. ↑ | Peak ↓ | Dep. ↓ | Acc. ↑ | Peak ↓ | Dep. ↓ |
| **Agent:** gpt-4.1-mini / **Compressor:** gpt-4.1-mini | | | | | | | | | | | | | |
| No Compression | 72.63 | 11.96 | 7.36 | 3.92 | 88.10 | 6.66 | 4.29 | 68.18 | 4.97 | 1.01 | 54.84 | 9.02 | 5.40 |
| **History Compression** | | | | | | | | | | | | | |
| FIFO | 65.26 | 10.91 | 4.03 | 1.46 | 83.33 | 4.10 | 0.78 | 59.09 | 3.69 | 0.96 | 45.16 | 4.19 | 2.03 |
| Retrieval | 67.37 | 14.46 | 4.55 | 2.74 | 85.71 | 5.85 | 5.86 | 59.09 | 3.47 | 0.87 | 48.39 | 4.59 | 2.45 |
| LLMLingua | 67.39 | 11.59 | 4.90 | 2.18 | 87.18 | 4.31 | 3.87 | 59.09 | 4.58 | 0.92 | 48.39 | 5.34 | 2.17 |
| Prompting | 71.58 | 11.78 | 4.93 | 3.10 | 85.71 | 4.73 | 4.75 | 72.73 | 4.40 | 0.86 | 51.61 | 5.32 | 3.06 |
| ACON | 73.68 | 12.41 | 4.82 | 1.96 | 88.10 | 4.12 | 0.83 | 68.18 | 4.39 | 0.86 | 58.06 | 5.37 | 3.07 |
| **Observation Compression** | | | | | | | | | | | | | |
| LLMLingua | 66.32 | 11.02 | 6.34 | 2.40 | 78.57 | 6.09 | 2.12 | 63.64 | 4.82 | 0.97 | 51.61 | 7.30 | 3.34 |
| Prompting | 73.68 | 11.43 | 6.45 | 2.62 | 88.10 | 4.82 | 1.44 | 72.73 | 4.95 | 1.06 | 54.84 | 8.01 | 4.01 |
| ACON | 71.58 | 10.96 | 6.00 | 2.19 | 88.10 | 4.45 | 1.06 | 63.64 | 4.89 | 1.00 | 54.84 | 7.30 | 3.36 |

*Table 11.* Results on **8-objective QA** benchmark with gpt-4.1-mini. We adopt the same compression guidelines as those used in the gpt-4.1 experiments. We report EM/F1 and efficiency metrics (Steps, Peak input tokens ($10^3$), and Dependency ($10^6$)).

| Method | EM ↑ | F1 ↑ | Steps ↓ | Peak ↓ | Dep. ↓ |
|---|---|---|---|---|---|
| **Agent:** gpt-4.1-mini / **Compressor:** gpt-4.1-mini | | | | | |
| No compression | 0.330 | 0.436 | 19.80 | 12.93 | 5.63 |
| **History Compression** | | | | | |
| FIFO | 0.024 | 0.031 | 28.45 | 5.33 | 3.89 |
| Retrieval | 0.143 | 0.190 | 26.90 | 5.34 | 3.55 |
| LLMLingua | 0.140 | 0.194 | 25.24 | 6.69 | 3.92 |
| Prompting | 0.149 | 0.207 | 25.27 | 4.85 | 2.44 |
| ACON | 0.238 | 0.325 | 21.05 | 4.78 | 2.03 |
| ACON (iter2) | 0.248 | 0.353 | 19.18 | 4.79 | 1.79 |
| **Observation Compression** | | | | | |
| LLMLingua | 0.316 | 0.430 | 15.96 | 5.54 | 1.60 |
| Prompting | 0.282 | 0.402 | 11.71 | 3.91 | 0.65 |
| ACON | 0.323 | 0.434 | 14.42 | 4.71 | 1.10 |
| ACON (iter2) | 0.316 | 0.443 | 11.69 | 3.97 | 0.63 |

*Table 12.* Results across different difficulty levels on **AppWorld** benchmark (test-normal) with gpt-5-chat. We adopt the same compression guidelines as those used in the gpt-4.1 experiments. Each block reports accuracy (task goal completion score), steps, peak input tokens ($10^3$), and dependency ($10^6$) for agents. Best results in each column are highlighted in bold. Rows in blue background indicate the results from ours .

| Method | Average (168) | | | | Easy (57) | | | Medium (48) | | | Hard (63) | | |
|---|---|---|---|---|---|---|---|---|---|---|---|---|---|
| | Acc. ↑ | Steps ↓ | Peak ↓ | Dep.↓ | Acc. ↑ | Peak ↓ | Dep.↓ | Acc. ↑ | Peak ↓ | Dep.↓ | Acc. ↑ | Peak ↓ | Dep.↓ |
| **Agent:** gpt-5-chat **/ Compressor:** gpt-5-chat | | | | | | | | | | | | | |
| No compression | 66.7 | 16.45 | 9.67 | 4.78 | 89.5 | 7.55 | 2.31 | 64.6 | 9.58 | 4.13 | 47.6 | 11.67 | 7.51 |
| **History Compression** | | | | | | | | | | | | | |
| FIFO (last-5) | 46.4 | 30.61 | 6.81 | 4.85 | 79.0 | 5.21 | 2.10 | 43.8 | 6.82 | 5.50 | 19.1 | 8.24 | 6.84 |
| Prompting | 58.9 | 22.24 | 7.46 | 4.02 | 82.5 | 7.15 | 2.13 | 66.7 | 7.19 | 3.69 | 31.8 | 7.93 | 5.97 |
| ACON UT | 58.3 | 20.15 | 6.97 | 3.74 | 80.7 | 6.66 | 2.04 | 66.7 | 7.08 | 3.40 | 31.8 | 7.16 | 5.54 |
| ACON UTCO | 62.5 | 22.29 | 7.26 | 3.85 | 86.0 | 6.44 | 2.04 | 72.9 | 6.98 | 3.93 | 33.3 | 8.20 | 5.42 |
| **Observation Compression** | | | | | | | | | | | | | |
| Prompting | 60.1 | 17.39 | 6.50 | 3.72 | 80.7 | 4.98 | 1.72 | 68.8 | 6.40 | 3.48 | 34.9 | 7.96 | 5.70 |
| ACON UT | 65.5 | 17.16 | 7.58 | 3.96 | 84.2 | 5.62 | 1.94 | 68.8 | 7.49 | 3.46 | 46.0 | 9.41 | 6.16 |
| ACON UTCO | 62.5 | 18.21 | 7.21 | 4.24 | 80.7 | 5.52 | 2.02 | 70.8 | 7.18 | 3.69 | 39.7 | 8.76 | 6.67 |
| **History + Observation Compression** | | | | | | | | | | | | | |
| ACON UT | 63.1 | 20.02 | 5.89 | 3.63 | 77.2 | 5.27 | 1.92 | 77.1 | 6.03 | 3.52 | 39.7 | 6.35 | 5.28 |
| ACON UTCO | 58.9 | 22.90 | 5.83 | 4.07 | 80.7 | 5.35 | 1.94 | 77.1 | 5.94 | 3.56 | 25.4 | 6.17 | 6.39 |

*Table 13.* Results across different difficulty levels on **AppWorld** with **distilled compressors**. We report accuracy (task goal completion score), average steps, peak input tokens ($10^3$), and dependency ($10^6$). For all compressors, we use the optimized compression guideline after the utilization maximization UT step. 'Fine-tune' means that we fine-tune small models with outputs from naive prompt before compression guideline optimization.

| Method | Average | | | | Easy | | | Medium | | | Hard | | |
|---|---|---|---|---|---|---|---|---|---|---|---|---|---|
| | Acc. ↑ | Steps ↓ | Peak ↓ | Dep.↓ | Acc. ↑ | Peak ↓ | Dep.↓ | Acc. ↑ | Peak ↓ | Dep.↓ | Acc. ↑ | Peak ↓ | Dep.↓ |
| **Agent:** gpt-4.1 **/ Compressor:** gpt-4.1-mini **or Distilled models** (Qwen3, Phi-4) | | | | | | | | | | | | | |
| **History Compression** | | | | | | | | | | | | | |
| Prompting (gpt-4.1-mini) | 39.3 | 23.61 | 7.03 | 5.19 | 64.9 | 6.64 | 3.17 | 35.4 | 7.63 | 5.42 | 19.1 | 6.93 | 6.84 |
| ACON (gpt-4.1-mini) | 47.6 | 21.46 | 7.25 | 5.24 | 75.4 | 6.75 | 2.84 | 35.4 | 7.25 | 5.36 | 31.8 | 7.70 | 7.32 |
| Fine-tune (Qwen3-14B) | 44.6 | 24.16 | 7.16 | 4.95 | 71.9 | 6.79 | 2.88 | 43.8 | 7.39 | 4.88 | 20.6 | 7.33 | 6.88 |
| ACON (Qwen3-14B) | 50.0 | 21.72 | 6.83 | 4.80 | 79.0 | 6.42 | 2.54 | 50.0 | 6.87 | 4.89 | 23.8 | 7.17 | 6.79 |
| ACON (Qwen3-8B) | 47.0 | 21.58 | 6.98 | 4.76 | 71.9 | 6.64 | 2.93 | 37.5 | 7.24 | 4.67 | 31.8 | 7.09 | 6.48 |
| ACON (Phi-4) | 44.6 | 21.19 | 7.24 | 4.76 | 68.4 | 7.33 | 2.75 | 39.6 | 7.12 | 4.16 | 27.0 | 7.26 | 7.04 |
| **Observation Compression** | | | | | | | | | | | | | |
| Prompting (gpt-4.1-mini) | 44.0 | 16.67 | 6.84 | 4.30 | 71.9 | 5.08 | 2.19 | 35.4 | 6.72 | 3.77 | 25.4 | 8.53 | 6.61 |
| ACON (gpt-4.1-mini) | 48.2 | 18.00 | 8.66 | 6.62 | 71.9 | 6.05 | 2.60 | 37.5 | 9.23 | 7.41 | 34.9 | 10.60 | 9.65 |
| Fine-tune (Qwen3-14B) | 40.5 | 17.71 | 6.64 | 4.38 | 64.9 | 4.91 | 1.97 | 31.2 | 6.72 | 4.05 | 25.4 | 8.16 | 6.81 |
| ACON (Qwen3-14B) | 56.5 | 16.78 | 7.57 | 5.06 | 82.5 | 5.69 | 2.20 | 54.2 | 7.39 | 4.46 | 34.9 | 9.40 | 8.10 |
| ACON (Qwen3-8B) | 48.2 | 16.10 | 7.33 | 4.82 | 71.9 | 5.49 | 2.03 | 50.0 | 7.20 | 4.20 | 25.4 | 9.10 | 7.82 |
| ACON (Phi-4) | 50.6 | 16.88 | 7.88 | 5.41 | 77.2 | 5.85 | 2.88 | 52.1 | 7.75 | 4.77 | 25.4 | 9.83 | 8.18 |

*Table 14.* Additional results for additional guideline optimization step and unified compression on **Appworld** benchmark (test-normal). Each block reports accuracy (task goal completion score), steps, peak input tokens ($10^3$), and dependency ($10^6$) for agents. Best results in each column are highlighted in bold. Rows in blue background indicate the results from ours .

| Method | Average (168) | | | | Easy (57) | | | Medium (48) | | | Hard (63) | | |
|---|---|---|---|---|---|---|---|---|---|---|---|---|---|
| | Acc. ↑ | Steps ↓ | Peak ↓ | Dep.↓ | Acc. ↑ | Peak ↓ | Dep.↓ | Acc. ↑ | Peak ↓ | Dep.↓ | Acc. ↑ | Peak ↓ | Dep.↓ |
| **Agent:** gpt-4.1 **/ Compressor:** gpt-4.1 | | | | | | | | | | | | | |
| **History Compression** | | | | | | | | | | | | | |
| ACON UTCOUT | 47.0 | 22.28 | 7.22 | 4.66 | 68.4 | 7.01 | 2.69 | 58.3 | 7.16 | 4.39 | 19.1 | 7.45 | 6.65 |
| **History + Observation Compression** | | | | | | | | | | | | | |
| Prompting | 36.3 | 19.33 | 5.38 | 3.44 | 71.9 | 4.87 | 1.80 | 21.6 | 5.63 | 3.60 | 14.3 | **5.64** | 4.79 |
| ACON | 45.8 | 20.32 | 5.85 | 4.26 | 75.4 | 5.29 | 2.07 | 39.6 | 6.15 | 4.29 | 23.8 | 6.12 | 6.21 |
| ACON UTCO | 44.6 | 21.75 | 5.90 | 4.98 | 77.2 | 5.50 | 2.33 | 39.6 | 6.18 | 3.80 | 19.1 | 6.18 | 8.28 |

---

**Prompt D.1: Prompt for analysis before prompt optimization (utility step)**

```
You are an expert agent trajectory auditor.

Analyze why the HISTORY-OPTIMIZED agent failed OR became significantly less
    efficient while the BASELINE succeeded.
You are given:
- task_name: {{ task_name }}
- Baseline full history (single continuous session)
- Optimized history split into multiple sessions where each new session starts
    with a fresh system + user prompt and an injected <HISTORY_SUMMARY>
    summarizing earlier interactions.
- baseline_success={{ baseline_success }} optimized_success={{ optimized_success
     }}
- baseline_env_steps={{ baseline_env_steps | default('null') }}
    optimized_env_steps={{ optimized_env_steps | default('null') }} step_ratio
    ={{ step_ratio | default('null') }} performance_regression={{
    performance_regression | default('false') }}

Goals:
1. Determine whether summarization / session resetting removed, distorted,
    delayed, or bloated reasoning causing failure OR inflated step count (>
    threshold factor of baseline).
2. Identify the FIRST divergence point where the optimized trajectory
    meaningfully deviates from the successful & efficient baseline path.
3. Categorize root causes (e.g., Missing Critical Fact, Incorrect Summary, Lost
    Variable/State, Unnecessary Re-discovery, Instruction Drift, API Misuse,
    Premature Completion, Token Truncation, Inefficient Looping, Redundant API
    Calls, Over-Exploration, Other).
4. Extract concrete evidence snippets (quote exact lines) from baseline vs
    optimized showing:
   - Critical facts present in baseline but absent/altered in optimized (esp.
    after a session boundary)
   - Summary inaccuracies (baseline ground truth vs summary text)
   - Redundant or looping action patterns causing step inflation.
5. Suggest precise remediation strategies: summary style changes, retain
    variable/value tables, move session boundaries, guardrail prompts, caching,
    early-exit heuristics, loop detection, etc.
6. Provide a reliability_score (0.0-1.0) reflecting confidence in your causal
    attribution.
7. If performance_regression==true, analyze efficiency degradation even if
    optimized_success==true.

Output STRICTLY valid JSON object with keys:
{
  "task_name": str,
  "divergence_step_description": str,
  "root_cause_categories": [str, ...],
  "missing_or_distorted_facts": [ {"baseline": str, "
    optimized_context_absent_or_changed": str, "impact": str} ],
  "summary_inaccuracies": [ {"summary_excerpt": str, "issue_type": str, "
    correct_baseline_reference": str, "impact": str} ],
  "lost_state_variables": [ {"name_or_pattern": str, "baseline_evidence": str, "
    optimized_issue": str} ],
  "api_or_action_errors": [ {"optimized_step_excerpt": str, "error_type": str, "
    improvement": str} ],
  "inefficiency_patterns": [ {"pattern": str, "evidence_excerpt": str, "
    excess_steps": int, "cause": str, "remediation": str} ],
  "timeline_of_divergence": [ {"phase": str, "optimized_excerpt": str, "
    baseline_contrast": str, "effect": str} ],
  "performance_regression": bool,
  "baseline_env_steps": int | null,
  "optimized_env_steps": int | null,
```

```
  "step_ratio": float | null,
  "remediation_recommendations": [ str, ... ],
  "recovery_opportunities_missed": [ {"optimized_excerpt": str, "
    missed_fix_action": str} ],
  "reliability_score": float,
  "concise_failure_mechanism_summary": str
}

If some sections have no data, use an empty list. For non-applicable numeric
    fields use null.
Do NOT include any extra commentary outside JSON.

---
BASELINE_HISTORY_START
{{ baseline_history }}
BASELINE_HISTORY_END

OPTIMIZED_MULTI_SESSION_HISTORY_START
{{ optimized_history }}
OPTIMIZED_MULTI_SESSION_HISTORY_END

Failure or performance report / metadata (may be null):
{{ failure_report }}

Proceed with rigorous comparison.
```

**Prompt D.2: Prompt for prompt optimization after analysis (utility step)**

```
You are an expert prompt engineer tasked with refining a HISTORY SUMMARIZATION
    prompt.
Rewrite the ORIGINAL PROMPT to reduce length of the HISTORY SUMMARY while
    preserving factual continuity for the next session.
Ground all changes in the PER-SAMPLE REDUCTION SIGNALS below. Do not aggregate
    across samples; use the patterns and rules as-is.

Constraints:
- Keep all Jinja placeholders, variable names, and structure intact where
    possible.
- Add explicit, concrete rules that prevent verbosity and retain essential state
    .
- Do not include literal values from prior content; refer to variable names only
    .
- Output ONLY the improved prompt template (no extra commentary).

Context (samples below are the only ground truth signals to use):
- Average original summary size (chars) across sampled set: {{ avg_orig_chars }}

{% for s in samples %}
===== SAMPLE {{ loop.index0 }} =====
- Task/Session: {{ s.task_label }} / {{ s.session or 'unknown-session' }}
- Analysis Overview:
{% if s.overview %}
{% for k, v in s.overview.items() %}  - {{ k }}: {{ v }}
{% endfor %}
{% else %}  - (none provided)
{% endif %}

- Removals (patterns -> action):
{% for r in s.removals %}  - [{{ r.category | default('unknown') }}] {{ r.
    pattern | default('') }} -> {{ r.action | default('drop') }}
{% endfor %}
```

```
- KEEP examples (evidence-driven essentials):
{% for k in s.keeps %}  - Reason: {{ k.reason | default('') }} | Evidence: {{ k.
    evidence_spans | default([]) | join('; ') }}
{% endfor %}

- Summary Rules:
{% for rule in s.rules %}  - {{ rule }}
{% endfor %}

{% endfor %}

Original Prompt Template (verbatim between markers):
<<<ORIGINAL_PROMPT>>>
{{ original_prompt }}
<<<ORIGINAL_PROMPT>>>

Output only the improved prompt template text, ready to be used as a Jinja
    template.
```

**Prompt D.3: Prompt for analysis before prompt optimization (compression step)**

```
You are an expert prompt engineer tasked with refining a HISTORY SUMMARIZATION
    prompt.
Rewrite the ORIGINAL PROMPT to reduce length of the HISTORY SUMMARY while
    preserving factual continuity for the next session.
Ground all changes in the PER-SAMPLE REDUCTION SIGNALS below. Do not aggregate
    across samples; use the patterns and rules as-is.

Constraints:
- Keep all Jinja placeholders, variable names, and structure intact where
    possible.
- Add explicit, concrete rules that prevent verbosity and retain essential state
    .
- Do not include literal values from prior content; refer to variable names only
    .
- Output ONLY the improved prompt template (no extra commentary).

Context (samples below are the only ground truth signals to use):
- Average original summary size (chars) across sampled set: {{ avg_orig_chars }}

{% for s in samples %}
===== SAMPLE {{ loop.index0 }} =====
- Task/Session: {{ s.task_label }} / {{ s.session or 'unknown-session' }}
- Analysis Overview:
{% if s.overview %}
{% for k, v in s.overview.items() %}  - {{ k }}: {{ v }}
{% endfor %}
{% else %}  - (none provided)
{% endif %}

- Removals (patterns -> action):
{% for r in s.removals %}  - [{{ r.category | default('unknown') }}] {{ r.
    pattern | default('') }} -> {{ r.action | default('drop') }}
{% endfor %}

- KEEP examples (evidence-driven essentials):
{% for k in s.keeps %}  - Reason: {{ k.reason | default('') }} | Evidence: {{ k.
    evidence_spans | default([]) | join('; ') }}
{% endfor %}
```

```
- Summary Rules:
{% for rule in s.rules %} - {{ rule }}
{% endfor %}

{% endfor %}

Original Prompt Template (verbatim between markers):
<<<ORIGINAL_PROMPT>>>
{{ original_prompt }}
<<<ORIGINAL_PROMPT>>>

Output only the improved prompt template text, ready to be used as a Jinja
    template.
```

---

**Prompt D.4: Prompt for analysis before prompt optimization (compression step)**

```
You are an expert prompt engineer tasked with refining a HISTORY SUMMARIZATION
    prompt.
Rewrite the ORIGINAL PROMPT to reduce length of the HISTORY SUMMARY while
    preserving factual continuity for the next session.
Ground all changes in the PER-SAMPLE REDUCTION SIGNALS below. Do not aggregate
    across samples; use the patterns and rules as-is.

Constraints:
- Keep all Jinja placeholders, variable names, and structure intact where
    possible.
- Add explicit, concrete rules that prevent verbosity and retain essential state
    .
- Do not include literal values from prior content; refer to variable names only
    .
- Output ONLY the improved prompt template (no extra commentary).

Context (samples below are the only ground truth signals to use):
- Average original summary size (chars) across sampled set: {{ avg_orig_chars }}

{% for s in samples %}
===== SAMPLE {{ loop.index0 }} =====
- Task/Session: {{ s.task_label }} / {{ s.session or 'unknown-session' }}
- Analysis Overview:
{% if s.overview %}
{% for k, v in s.overview.items() %} - {{ k }}: {{ v }}
{% endfor %}
{% else %} - (none provided)
{% endif %}

- Removals (patterns -> action):
{% for r in s.removals %} - [{{ r.category | default('unknown') }}] {{ r.
    pattern | default('') }} -> {{ r.action | default('drop') }}
{% endfor %}

- KEEP examples (evidence-driven essentials):
{% for k in s.keeps %} - Reason: {{ k.reason | default('') }} | Evidence: {{ k.
    evidence_spans | default([]) | join('; ') }}
{% endfor %}

- Summary Rules:
{% for rule in s.rules %} - {{ rule }}
{% endfor %}

{% endfor %}
```

```
Original Prompt Template (verbatim between markers):
<<<ORIGINAL_PROMPT>>>
{{ original_prompt }}
<<<ORIGINAL_PROMPT>>>

Output only the improved prompt template text, ready to be used as a Jinja
    template.
```

---

**Prompt D.5: AppWorld Prompt for history compression before optimization**

```
You are maintaining a structured context-aware summary for a productivity agent.
     You will be given the user instruction for the agent, a list of
    interactions corresponding to actions taken by the agent, and the most
    recent previous summary if one exists. Produce the following:

### REASONING
Summarize key progress, decisions made, important observed outcomes, and
    rationale behind actions taken so far. Include how earlier steps influenced
    later ones and why certain data is retained in the summary.

### COMPLETED
List completed subtasks or successful outcomes, with brief results if applicable
    .

---

## [Information Source]

### USER INSTRUCTION

{{ task }}

## [PREVIOUS SUMMARY] (if any)

{{ prev_summary }}

## [HISTORY OF INTERACTIONS]

{{ history }}

---

## PRIORITIZE

1. Keep all sections relevant and concise.
2. Use reusable structured formats when summarizing artifacts.
3. Ensure agent can resume task with no loss of information.
4. Include key info from errors or failed attempts to prevent repeated mistakes.
5. Preserve all essential artifacts and data needed to complete the task.

---

### [Output Format]

Do **not** include the input or any additional explanation. Only return the
    formatted summary.
```

**Prompt D.6: AppWorld Prompt for history compression after optimization (UT)**

```
You maintain a compact, state-preserving HISTORY_SUMMARY for a multi-session
    agent.

Input:
[USER INSTRUCTION] {{ task }}
[PREVIOUS SUMMARY] {{ prev_summary }}
[HISTORY OF INTERACTIONS] {{ history }}

Create the following sections-use the exact headings and order:

<HISTORY_SUMMARY>

1. REASONING
   - Key progress, decisions, outcomes, and their rationale.
   - Note how earlier steps influence later ones.

2. VARS
   | name | value | purpose |
   |------|-------|---------|
   Record every runtime value the next session must re-declare (tokens, ids,
    lists, last page_index/page_limit, etc.).

3. TODO
   List pending actions with enough detail to execute directly.

4. COMPLETED
   Bullet list of finished subtasks with brief results.

5. GUARDRAILS
   Short reminders that prevent repeat errors, e.g.
   - Memory resets; re-create VARS before use.
   - Paginate until empty page.
   - Validate API parameters against spec.
   - Avoid redundant logins or doc look-ups.

Requirements:
- Be concise-bullets and tables preferred; no extraneous prose.
- Preserve all essential facts, parameters, and artifacts; omit nothing critical
    .
- Include errors only if they inform future avoidance.
- Do not output the input or any commentary-return only <HISTORY_SUMMARY>.
```

**Prompt D.7: AppWorld Prompt for history compression after optimization (UT CO)**

```
You maintain a compact, state-preserving HISTORY_SUMMARY for a multi-session
    agent.

Input:
[USER INSTRUCTION] {{ task }}
[PREVIOUS SUMMARY] {{ prev_summary }}
[HISTORY OF INTERACTIONS] {{ history }}

Summary Compression Rules:
- Collapse multi-bullet narratives into <=2 concise sentences.
- Replace repetitive step logs with one summarizing phrase.
- Truncate long token/credential strings to "<token>" unless verbatim reuse is
    required.
```

```
- Remove unused/expired credentials, page_index/page_limit, verbose API dumps,
    and table borders.
- Shrink GUARDRAILS to one bullet unless multiple items are still critical.
- Delete tool/API log output, greetings, meta prose, and section headers that no
    longer contain content.
- Keep only variables actively referenced in upcoming steps; list each once in
    VARS.
- Reference removal categories [repetition], [tool-logs], [meta], [formatting]
    to prune similar lines.
- Preserve factual continuity; never invent or alter state variables.
- Target summaries well under {{ max_chars | default(1500) }} characters.

Critical Essentials:
Always keep evidence-driven items required next session (e.g., tokens, ids,
    emails, amounts, lists, paths, description strings, brief task status).

Output EXACTLY the following structure---nothing more:

<HISTORY_SUMMARY>

1. REASONING
    One brief paragraph on key progress and rationale.

2. VARS
    key=value pairs, comma-separated; only still-needed runtime values.

3. TODO
    Bulleted next actions (<=5).

4. COMPLETED
    Bulleted finished subtasks (<=5).

5. GUARDRAILS
    Single concise bullet, or omit if none.

Return only the <HISTORY_SUMMARY> block---no additional commentary or input
    echoes.
```

---

**Prompt D.8: AppWorld Prompt for observation compression before optimization**

```
Your task is to generate a "Reasoning" and a "Refined Observation" based on the
    inputs below.

In the "Reasoning", analyze the user instruction and history to identify what
    information from the current observation is necessary to complete the
    remaining steps.
Think about what parts can be summarized or transformed to reduce length, while
    ensuring that future actions can still be executed based on the refined
    observation alone.

In the "Refined Observation", include only the information that is minimal but
    sufficient for the next steps.

[Information source]
# User Instruction
{{ task }}

# History of interactions
{{ history }}

# Observation at the current time step
```

```
{{ observation }}

[Output format]
# Reasoning
... your reasoning for what matters and how to optimize it ...
# Refined Observation
... reduced and actionable observation ...
```

**Prompt D.9: AppWorld Prompt for observation compression after optimization (UT)**

```
Your task: write two sections---"Reasoning" and "Refined Observation".

1. Reasoning
   - Examine task, history, and observation.
   - Decide exactly which parts of the observation must be kept so the next
    agent step can succeed.
   - Note any need to paginate (page_limit default = 5, page_index).
   - Justify any data you drop.

2. Refined Observation
   - Contain only the minimal yet sufficient info for the next step.
   - Always preserve:
     - Every endpoint that may be called, plus its full parameter list and
    defaults (especially page_limit/page_index, auth tokens).
     - Response-schema fields referenced or likely needed later (e.g.,
    play_count, release_date, like_count, position, ids).
     - Raw data rows required for future comparisons or loops; if summarising,
    keep at least all positive-match examples.
   - Never:
     - Omit defaults that affect behaviour.
     - Declare parameters "not critical" without proof.
     - Hallucinate endpoints or fields.
     - Replace machine-readable data with vague prose.

[Information source]
# User Instruction
{{ task }}

# History of interactions
{{ history }}

# Observation at the current time step
{{ observation }}

[Output format]
# Reasoning
...concise analysis explaining what is kept/removed...

# Refined Observation
...trimmed yet complete observation ensuring future steps remain possible...
```

**Prompt D.10: AppWorld Prompt for observation compression after optimization (UTCO)**

```
Your task: create two sections---"Reasoning" and "Refined Observation".

1. Reasoning (<=40 words)
```

```
    – Briefly state what was kept and why; note dropped categories and any
     pagination needs.

2. Refined Observation (use ONLY the current observation)
    – Keep strictly necessary data for the next step.
    – Always preserve, when present:
      – access_token or other auth values
      – page_limit, page_index and other defaults that affect calls
      – every endpoint name; include parameters only if required; description <=
     4 words
      – identifiers/fields needed for comparisons, loops, or API calls (ids,
     titles, counts, paths, etc.)
    – Minimise length:
      – Delete unused fields and narrative text.
      – Compress JSON/arrays (one object per line, no extra spaces).
      – Summarise long uniform lists with a range/pattern when individual rows
     are not needed.
    – Never invent, alter, or omit a required literal.
    – Exclude history, prior summaries, and meta comments.

[Information source]
# User Instruction
{{ task }}

# History of interactions
{{ history }}

# Observation at the current time step
{{ observation }}

[Output format]
# Reasoning
...

# Refined Observation
...
```

---

**Example D.1: AppWorld task trajectory example**

**Problem:** Name the artist least recommended to me on Spotify.

---

```python
# First, let's check what APIs are available under the Spotify app to see if there is
↪   an API related to recommendations or artists.
print(apis.api_docs.show_api_descriptions(app_name='spotify'))
```

```
[{"name":"login",...},{"name":"show_recommendations","description":"Get
↪   personalized song recommendations for the user."}, ...]
```

```python
# There is an API called "show_recommendations" which provides personalized song
↪   recommendations. Let's check its detailed specification...
print(apis.api_docs.show_api_doc(app_name='spotify', api_name='show_recommendations'))
```

```
{"path":"/recommendations","method":"GET","parameters":[{"name":"access_token","r⌋
↪   equired":true}, {"name":"page_index",...},{"name":"page_limit",...}], ...}
```

```python
# To call `show_recommendations`, I need an access token. Check the login spec.
print(apis.api_docs.show_api_doc(app_name='spotify', api_name='login'))
```

```
{"path":"/auth/token","method":"POST","parameters":[{"name":"username"},{"name":"⌋
↪   password"}], ...}
```

```python
# I need Timothy's Spotify account password to log in. Let's get it from supervisor.
spotify_passwords = apis.supervisor.show_account_passwords()
spotify_password = next((acc["password"] for acc in spotify_passwords if
↪   acc["account_name"]=="spotify"), None)
print(spotify_password)
```

```
<REDACTED>
```

```python
# Now log in to get the access token.
spotify_login = apis.spotify.login(username="timothy.whit@gmail.com",
                                   password="<REDACTED>")
access_token = spotify_login.get("access_token")
print(access_token)
```

```
eyJhbGciOiJIUzI1NiIsInR5cCI6IkpXVCJ9...<REDACTED>...
```

```python
# Now get the first page of personalized song recommendations.
recommendations_page_0 = apis.spotify.show_recommendations(
    access_token=access_token, page_index=0, page_limit=20
)
print(recommendations_page_0[:4])  # show a few items only
```

```
[
 {"song_id":10,"title":"The Curse of Loving You","artists":[{"name":"Lucas
 ↪   Grey"}]},
 {"song_id":11,"title":"Lost in a Moment's Grace","artists":[{"name":"Apollo
 ↪   Serenade"},{"name":"Phoenix Rivers"}]},
 {"song_id":40,"title":"Burning Like a Distant Star","artists":[{"name":"Aria
 ↪   Sterling"}]},
 ...
]
```

```
# To find the artist least recommended to Timothy, I will count how many times each
↪   artist appears in the recommendations. The artist with the fewest appearances is
↪   the least recommended.
from collections import Counter

artist_names = [a["name"] for r in recommendations_page_0 for a in r["artists"]]
artist_counts = Counter(artist_names)
least_recommended_artist = min(artist_counts, key=artist_counts.get)
print(least_recommended_artist)
```

```
  Evelyn Rose
```

```
# Task complete. I will submit "Evelyn Rose" as the answer.
apis.supervisor.complete_task(answer="Evelyn Rose")
```

**Example D.2:** AppWorld trajectory example - gpt-4.1-mini agent fails without history compression

**Problem:** Delete all `.pdf` files from `~/downloads`.

---

```
# Understand available APIs in the file_system app.

print(apis.api_docs.show_api_descriptions(app_name='file_system'))
```

```
  [
  {"name":"show_account","description":"Show your account information. Unlike
  ↪   show_profile, this includes private information."},
  {"name":"signup","description":"Sign up to create account."},
  {"name":"delete_account","description":"Delete your account."},
  {"name":"update_account_name","description":"Update your first or last name in the
  ↪   account profile."},
  ...
  {"name":"copy_directory","description":"Copy a directory to another location."},
  {"name":"move_directory","description":"Move a directory to another location."},
  {"name":"compress_directory","description":"Compress a directory."},
  {"name":"decompress_file","description":"Decompress a compressed file."}
  ]
```

```
# List files in ~/downloads to find PDFs.

downloads_path = '~/downloads'
downloads_files = apis.file_system.show_directory(path=downloads_path)
print(downloads_files)
```

```
  Execution failed. Traceback:
  ...
  Exception: Response status code is 401:
  {"message":"You are either not authorized to access this file_system API endpoint
  ↪   or your access token is missing, invalid or expired."}
```

```
# Need authentication. Check supervisor for stored credentials.

print(apis.supervisor.show_account_passwords())
```

```
[
{"account_name":"amazon","password":"Gt$!_*W"},
{"account_name":"file_system","password":"8nNw!jZ"},
{"account_name":"gmail","password":"qu4Y7}s"},
{"account_name":"phone","password":"C4n&I40"},
{"account_name":"simple_note","password":"e+QwbmV"},
{"account_name":"splitwise","password":"mSqG}QU"},
{"account_name":"spotify","password":"%CCvl8v"},
{"account_name":"todoist","password":"jHZ#RPM"},
{"account_name":"venmo","password":"Wq8!RAU"}
]
```

```
# Try logging in with just the password (no username).

file_system_password = [a["password"] for a in apis.supervisor.show_account_passwords()
if a["account_name"] == "file_system"][0]
file_system_login_response = apis.file_system.login(password=file_system_password)
print(file_system_login_response)
```

```
Execution failed. Traceback:
...
Exception: Response status code is 422:
{"message":"Validation error. Reason: \nusername: Field required"}
```

```
# Need username. Check supervisor profile for email/username.

print(apis.supervisor.show_profile())
```

```
{"first_name":"Susan","last_name":"Burton","email":"[susanmiller@gmail.com](mailt
⌋  o:susanmiller@gmail.com)",
"phone_number":"3296062648","birthday":"1994-04-30","sex":"female"}
```

```
# Attempt login with email as username.

file_system_username = "[susanmiller@gmail.com](mailto:susanmiller@gmail.com)"
file_system_password = [a["password"] for a in apis.supervisor.show_account_passwords()
if a["account_name"] == "file_system"][0]
file_system_login_response = apis.file_system.login(
username=file_system_username, password=file_system_password)
print(file_system_login_response)
```

```
{"access_token":"eyJhbGciOiJIUzI1NiIsInR5cCI6IkpXVCJ9...<REDACTED>...","token_typ
⌋  e":"Bearer"}
```

```
# Try to set token manually (guessing an API), then list directory.

apis.file_system.set_access_token("...<REDACTED>...")
downloads_path = '~/downloads'
downloads_files = apis.file_system.show_directory(path=downloads_path)
print(downloads_files)
```

```
Execution failed. Traceback:
...
Exception: No API named 'set_access_token' found in the file_system app.
```

```python
# Call show_directory again, assuming session is persisted after login.

downloads_path = '~/downloads'
downloads_files = apis.file_system.show_directory(path=downloads_path)
print(downloads_files)
```

```
  Execution failed. Traceback:
  ...
  Exception: Response status code is 401:
  {"message":"You are either not authorized to access this file_system API endpoint
  ↪  or your access token is missing, invalid or expired."}
```

```python
# Retry: fresh login, try immediate call.

login_response = apis.file_system.login(
username=file_system_username, password=file_system_password)
access_token = login_response.get("access_token")
print(access_token)  # for debugging
```

```
  eyJhbGciOiJIUzI1NiIsInR5cCI6IkpXVCJ9...<REDACTED>...
```

```python
# Try show_directory once more after login.

downloads_path = '~/downloads'
downloads_files = apis.file_system.show_directory(path=downloads_path)
print(downloads_files)
```

```
  Execution failed. Traceback:
  ...
  Exception: Response status code is 401:
  {"message":"You are either not authorized to access this file_system API endpoint
  ↪  or your access token is missing, invalid or expired."}
```

```python
# Inspect file_system APIs again for any token/session helpers.

print(apis.api_docs.show_api_descriptions(app_name='file_system'))
```

```
  [{"name":"show_account","description":"Show your account information. Unlike
  ↪  show_profile, this includes private information."},
  {"name":"signup","description":"Sign up to create account."},
  {"name":"delete_account","description":"Delete your account."},
  {"name":"update_account_name","description":"Update your first or last name in the
  ↪  account profile."},
  {"name":"login","description":"Login to your account."},
  ...
  {"name":"copy_directory","description":"Copy a directory to another location."},
  {"name":"move_directory","description":"Move a directory to another location."},
  {"name":"compress_directory","description":"Compress a directory."},
  {"name":"decompress_file","description":"Decompress a compressed file."}]
```

```python
# Try logout -> login -> call again.

apis.file_system.logout()
```

```
login_response = apis.file_system.login(
username=file_system_username, password=file_system_password)
downloads_path = '~/downloads'
downloads_files = apis.file_system.show_directory(path=downloads_path)
print(downloads_files)
```

```
   Execution failed. Traceback:
   ...
   Exception: Response status code is 422:
   {"message":"You are either not authorized to access this file_system API endpoint
   ↪  or your access token is missing, invalid or expired."}
```

```
# Final retry without logout; still unauthorized.

login_response = apis.file_system.login(
username=file_system_username, password=file_system_password)
downloads_path = '~/downloads'
downloads_files = apis.file_system.show_directory(path=downloads_path)
print(downloads_files)
```

```
   Execution failed. Traceback:
   ...
   Exception: Response status code is 401:
   {"message":"You are either not authorized to access this file_system API endpoint
   ↪  or your access token is missing, invalid or expired."}
```

```
# Supervisor cannot manage files; mark task failed due to authorization issues.

print(apis.api_docs.show_api_descriptions(app_name='supervisor'))
```

```
   [
   {"name":"show_active_task","description":"Show the currently active task assigned
   ↪  to you by the supervisor."},
   {"name":"complete_task","description":"Mark the currently active task as complete
   ↪  with the given answer."},
   {"name":"show_profile","description":"Show your supervisor's profile
   ↪  information."},
   {"name":"show_addresses","description":"Show your supervisor's addresses."},
   {"name":"show_payment_cards","description":"Show your supervisor's
   ↪  payment_cards."},
   {"name":"show_account_passwords","description":"Show your supervisor's app account
   ↪  passwords."}
   ]
```

```
apis.supervisor.complete_task(status="fail")
```

---

**Example D.3: AppWorld trajectory example - gpt-4.1-mini agent successes with history compression**

**Problem:** Delete all `.pdf` files from `~/downloads`.

---

```
(... previous raw history is omitted for brevity ...)
<HISTORY_SUMMARY>

1. REASONING
   - User requested deletion of all .pdf files in ~/downloads.
   - file_system APIs require authentication; initial calls failed with 401
   ↪  Unauthorized.
   (... omitted ...)
   - Confirmed login API returns access_token but no mechanism to use it in
   ↪  subsequent calls found.
   - Need to find a way to authorize file_system API calls with the access token
   ↪  or alternative approach.

2. VARS
   | name                | value                                         | purpose
   ↪  |
   |---------|----------|--------|
   | file_system_username | "susanmiller@gmail.com"                      | Username for
   ↪  file_system login  |
   | file_system_password | "8nNw!jZ"                                    | Password for
   ↪  file_system login  |
   | file_system_token   | "eyJhbGciOiJIUzI1NiIsInR5cCI6IkpXVCJ9...sBA8nTOB..." |
   ↪  Access token from login |
   | downloads_path      | "~/downloads"                                 | Target
   ↪  directory for file listing |

3. TODO
   - Determine how to pass or set the access token for file_system API calls to
   ↪  authorize them.
   - Once authorized, list files in ~/downloads to identify .pdf files.
   - Delete all .pdf files found in ~/downloads.

4. COMPLETED
   - Retrieved file_system account password from supervisor app.
   - Successfully logged in to file_system app with username and password,
   ↪  obtained access token.
   - Confirmed file_system APIs require authorization; calls without token fail.
   - Verified no supervisor app API to proxy file system operations.
   - Confirmed login API returns access token but no visible method to use it in
   ↪  subsequent calls.

5. GUARDRAILS
   - Always provide both username and password for login API.
   - Access tokens must be explicitly handled or passed if environment does not
   ↪  auto-use them.
   - Validate authorization before calling protected APIs to avoid 401 errors.
   - Avoid redundant login attempts without token usage resolution.
   - Preserve tokens and credentials securely for session continuity.
</HISTORY_SUMMARY>
```

```python
# The show_directory API requires the access_token as a parameter.

# List files in "~/downloads" using the token.

downloads_path = "~/downloads"
file_system_access_token = "eyJhbGciOiJIUzI1NiIsInR5cCI6IkpXVCJ9...sBA8nTOB..."

print(apis.file_system.show_directory(
access_token=file_system_access_token,
directory_path=downloads_path,
```

```
entry_type="files",
recursive=False))
```

```
  [
  "/home/susan/downloads/aerobics_class.jpg",
  "/home/susan/downloads/art_inspiration_sketches.zip",
  "/home/susan/downloads/bakeware.pdf",
  "/home/susan/downloads/bowls_for_snacks.pdf",
  "/home/susan/downloads/candy.pdf",
  "/home/susan/downloads/charcoal_pencils.pdf",
  ...,
  "/home/susan/downloads/travel_insurance.pdf",
  "/home/susan/downloads/workout_clothes.pdf",
  "/home/susan/downloads/world_cuisine_recipe_book.epub",
  "/home/susan/downloads/world_landmarks_photo_album.zip"
  ]
```

```
# Filter to only ".pdf" files.

downloads_files = [
"/home/susan/downloads/aerobics_class.jpg",
"/home/susan/downloads/bakeware.pdf",
"/home/susan/downloads/bowls_for_snacks.pdf",
"/home/susan/downloads/candy.pdf",
"/home/susan/downloads/charcoal_pencils.pdf",
...,
"/home/susan/downloads/travel_insurance.pdf",
"/home/susan/downloads/workout_clothes.pdf"
]
pdf_files = [p for p in downloads_files if p.lower().endswith(".pdf")]
print(pdf_files)
```

```
  [
  "/home/susan/downloads/bakeware.pdf",
  "/home/susan/downloads/bowls_for_snacks.pdf",
  "/home/susan/downloads/candy.pdf",
  "/home/susan/downloads/charcoal_pencils.pdf",
  ...,
  "/home/susan/downloads/travel_insurance.pdf",
  "/home/susan/downloads/workout_clothes.pdf"
  ]
```

```
# Confirm delete_file API parameters before deletion.

print(apis.api_docs.show_api_doc(app_name="file_system", api_name="delete_file"))
```

```
    {
    "app_name":"file_system",
    "api_name":"delete_file",
    "path":"/file",
    "method":"DELETE",
    "description":"Delete a file.",
    "parameters":[
    {"name":"file_path","type":"string","required":true,
    "description":"Path of the file. Path can be absolute, starting with '/', or
    ↪  relative to the user's home directory, starting with '~/'."},
    {"name":"access_token","type":"string","required":true,
    "description":"Access token obtained from file_system app login."}
    ],
    "response_schemas":{
    "success":{"message":"string","file_path":"string"},
    "failure":{"message":"string"}
    }
    }
```

```python
# Delete each PDF using the access_token.

delete_results = []
for pdf in pdf_files:
res = apis.file_system.delete_file(
file_path=pdf, access_token=file_system_access_token)
delete_results.append(res)

# Mark task complete.

apis.supervisor.complete_task()
```

---

**Compression D.1: AppWorld History Compression Example**

---

```
task: I got some Splitwise group invitations over phone text messages yesterday.
If their number is in my phone contact book, accept it, otherwise delete those
    messages.

----------------------------------------

History Compression (Prompting baseline):
### REASONING

The agent began by identifying relevant apps for handling text messages and
    Splitwise invitations, then explored the phone app's APIs to manage messages
    and contacts. To search for Splitwise group invitations received via text
    message, the agent determined it needed to log in to the phone app to obtain an
     access token. Using credentials from the supervisor app, the agent
    successfully logged in and retrieved the access token. The agent then
    calculated yesterday's date range and searched for text messages containing "
    Splitwise," retrieving a list of messages. The rationale for retaining message
    details (sender name, phone number, contact_id, message content, sent_at) is to
     enable the next step: checking if the sender's number is in the contact book,
    so the agent can accept or delete invitations as instructed. Earlier steps (API
     exploration, login, date calculation) directly enabled the message search and
    data extraction.

### COMPLETED

- Identified relevant apps and APIs for the task.
- Successfully logged in to the phone app and obtained access token.
- Calculated yesterday's date range.
- Searched and retrieved text messages mentioning "Splitwise" from yesterday and
    beyond, including sender details and message content.

----------------------------------------

History Compression (ACON (utility step)):
1. REASONING
   - Identified "phone" app for managing text messages and contacts.
   - Retrieved API docs to find endpoints for searching messages and contacts.
   - Determined yesterday's date (May 17, 2023) for filtering relevant messages.
   - Checked "search_text_messages" API spec for required parameters.
   - Used supervisor app to obtain phone app password.
   - Logged in to phone app to get access_token.
   - Searched for messages containing "Splitwise" with page_limit=20, sorted by -
    created_at.
   - Located Splitwise group invitation messages; sender contact_id used to
    determine if sender is in contact book.

2. VARS
   | name          | value
         | purpose                             |
   |---------------|---------------------------|------------------|
   | access_token  | eyJhbGciOiJIUzI1NiIsInR5cCI6IkpXVCJ9.
   eyJzdWIiOiJwaG9uZSs3OTM0ODE4MTkxIiwiZXhwIjoxNjg0NDEyMDk4fQ.
   ad3t3S_Xj7YhoDDFZeW4BlVL4dNniMdfaXC1143fbzw | Auth for phone app APIs
         |
   | phone_number  | 7934818191
         | User's phone number                 |
   | phone_password| X53y6aW
         | Phone app password                  |
   | search_query  | "Splitwise"
         | Filter for relevant messages        |
```

```
        | sort_by       | -created_at
             | Get latest messages first              |
        | page_index    | 0
             | Current page index for pagination      |
        | page_limit    | 20
             | Number of results per page             |
        | target_date   | 2023-05-17
             | Date to filter messages                |
        | invitation_msgs | [IDs: 16794, 16800]
             | Splitwise invitations from 2023-05-17     |
        | sender_info   | [{id:1857, name:"Alex White", phone:"3707865891"}, {id:null,
        name:"Anita Burch", phone:"3643463570"}] | For contact book check
             |
```

3. TODO
   - For each Splitwise invitation message from 2023-05-17:
     - Check if sender's phone number exists in contact book (search_contacts API).
     - If in contact book, accept invitation (Splitwise API, if needed).
     - If not in contact book, delete the message (delete_text_message API).
   - Paginate search_text_messages if more relevant messages may exist.

4. COMPLETED
   - Identified relevant apps and APIs for task.
   - Retrieved phone app password from supervisor.
   - Logged in to phone app and obtained access_token.
   - Searched and filtered text messages for Splitwise invitations on target date.
   - Isolated relevant message IDs and sender info for next steps.

5. GUARDRAILS
   - Always re-declare VARS after memory reset.
   - Paginate message search until no more results.
   - Validate API parameters before each call.
   - Avoid redundant logins and API doc look-ups.
   - Only act on messages matching both query and target date.

--------------------------------------

History Compression (ACON (utility step + compression step)):

1. REASONING
   I identified voice messages from the day before yesterday (2023-05-16)
   containing Splitwise group invitations. To follow instructions, I need to check
   if the senders' phone numbers are in Sierra's contact book before accepting or
   deleting these messages.

2. VARS
   access_token=<token>, splitwise_invite_voice_ids=[17022,17036,17028,17034],
   sender_numbers=['9595488531','7375474944','2123782445','3808719492']

3. TODO
   - Search Sierra's contact book for sender_numbers.
   - Accept Splitwise invitations from contacts found.
   - Delete voice messages from non-contacts.

4. COMPLETED
   - Logged in to phone app and obtained access_token.
   - Retrieved voice messages from 2023-05-16.
   - Identified Splitwise invitation messages and sender numbers.

5. GUARDRAILS
   Only process Splitwise invitations from contacts; delete others.

**Compression D.2: 8-objective QA History Compression Example**

```
task: when was the last time the dodgers played yankees in the world series?; where
     will be the next olympics be held?; who is tinker air force base named after?;
      who does the voice of cruz on cars 3?; who's the original singer of help me
    make it through the night?; where does the movie mothers day take place?; what
    is the big gold dome in jerusalem?; when did russia join the world economic
    forum?

----------------------------------------

History Compression (Prompting baseline):
### REASONING
The agent began by addressing the user's multi-part query, starting with "when was
    the last time the Dodgers played Yankees in the World Series?" Through
    Wikipedia searches, the agent identified 1981 as the most recent World Series
    meeting between the Dodgers and Yankees. This answer was retained due to clear
    evidence in the retrieved documents. The agent then attempted to answer "where
    will be the next Olympics be held?" but initial searches returned irrelevant or
     outdated results (e.g., Amsterdam 1928, Buenos Aires 2018). Recognizing the
    lack of direct answers, the agent refined its search queries multiple times,
    but still did not retrieve the current location for the next Olympics. The
    agent's iterative approach demonstrates a commitment to accuracy and highlights
     the need to adjust search strategies when initial attempts fail. Retaining
    details of failed searches helps prevent repeated mistakes and guides future
    queries.

### COMPLETED
- Identified the last time the Dodgers played the Yankees in the World Series:
    1981.

----------------------------------------

History Compression (ACON (utility step)):

### REASONING
- Used Wikipedia search for each question; refined Olympics query after initial
    results lacked direct info.
- Collapsed repeated/irrelevant Olympic bid results; focused on confirmed host city
    .
- For Tinker AFB, confirmed name origin from first document.

### COMPLETED
- Last Dodgers vs Yankees World Series: 1955 (source: "1955 World Series")
- Next Olympics location: Paris, 2024 (source: "Budapest bid for the 2024 Summer
    Olympics")
- Tinker Air Force Base named after Major General Clarence L. Tinker (source: "
    Tinker Air Force Base")

### STATE LOG
FACT CACHE:
- When was the last time the Dodgers played Yankees in the World Series? -> 1955
    ("1955 World Series")
- Where will the next Olympics be held? -> Paris, 2024 ("Budapest bid for the 2024
    Summer Olympics")
- Who is Tinker Air Force Base named after? -> Major General Clarence L. Tinker ("
    Tinker Air Force Base")

PENDING:
- Who does the voice of Cruz on Cars 3?
- Who is the original singer of "Help Me Make It Through the Night"?
- Where does the movie "Mother's Day" take place?
- What is the big gold dome in Jerusalem?
```

```
- When did Russia join the World Economic Forum?

ATTEMPTED QUERIES / LOOP GUARD:
- "Dodgers Yankees World Series history" OK
- "next Olympics location" FAIL (no direct info)
- "2024 Olympics location" OK
- "Tinker Air Force Base name origin" OK
- "next Olympics location" tried twice (loop guard: must reformulate/abandon)

ERROR NOTES:
None

(End of summary - no extra text)

----------------------------------------

History Compression (ACON (utility step + compression step)):

### FACT CACHE
- When was the last time the Dodgers played Yankees in the World Series? -> 1956

### PENDING
where will the next olympics be held?, who is tinker air force base named after?,
    who does the voice of cruz on cars 3?, who is the original singer of help me
    make it through the night?, where does the movie mothers day take place?, what
    is the big gold dome in jerusalem?, when did russia join the world economic
    forum?

### BLOCKED
next Olympics location: 3x failed searches (loop-guard)

### REASONING
Olympics location queries returned irrelevant results; alternate sources or query
    phrasing needed.
```

## Compression D.3: OfficeBench History Compression Example

```
task: read company revenues, save the top 2 revenue years in top2.xlsx, generate
    reports.docx

----------------------------------------

History Compression (Prompting baseline):
### REASONING

The agent began by identifying the relevant data file (company_revenues.xlsx)
    containing company revenue information. After reading the file, the agent
    determined the top 2 revenue years: 2014 ($4,983,791) and 2015 ($4,611,210).
    The agent then created a new Excel file (top2.xlsx) and started populating it
    with the required data, first adding headers ("Year", "Revenue ($)") and then
    entering the first top year (2014) in the second row, first column. The
    rationale for retaining row and column indices, file paths, and the top 2 years
     is to ensure accurate and complete transfer of data and to facilitate the next
     steps (completing the top2.xlsx file and generating reports.docx).

### COMPLETED

- Located and read company_revenues.xlsx.
- Identified top 2 revenue years: 2014 ($4,983,791), 2015 ($4,611,210).
- Created top2.xlsx and added headers ("Year", "Revenue ($)").
- Entered "2014" in (2, 1) of top2.xlsx.
```

```
----------------------------------------

History Compression (ACON (utility step)):

### REASONING
- Located company revenue data in company_revenues.xlsx.
- Read all rows; identified top 2 revenue years: 2014 ($4,983,791) and 2015 ($4
    ,611,210).
- Created top2.xlsx and began writing header and first top year.
- Each numeric value was copied verbatim from source.
- No errors or mismatches so far; headers and first year written successfully.
- Pending: finish writing revenue for 2014, add 2015 row, then generate reports.
    docx.

Verify numeric values directly from source file before use.

---

### DATA STATE

#### company_revenues.xlsx (source table)
| Row | Year | Revenue ($) |
|-----|------|-------------|
| 2   | 2004 | 4439044     |
| 3   | 2005 | 2212203     |
| 4   | 2006 | 1193225     |
| 5   | 2007 | 2793265     |
| 6   | 2008 | 2080460     |
| 7   | 2009 | 1791112     |
| 8   | 2010 | 3098049     |
| 9   | 2011 | 3975966     |
| 10  | 2012 | 3251777     |
| 11  | 2013 | 2918667     |
| 12  | 2014 | 4983791     |
| 13  | 2015 | 4611210     |
| 14  | 2016 | 3294631     |
| 15  | 2017 | 1463500     |
| 16  | 2018 | 3945846     |
| 17  | 2019 | 3907271     |
| 18  | 2020 | 3651262     |
| 19  | 2021 | 2005820     |
| 20  | 2022 | 2170828     |
| 21  | 2023 | 2832545     |

#### top2.xlsx (current state)
| (row, col) | Value          |
|------------|----------------|
| (1, 1)     | Year           |
| (1, 2)     | Revenue ($)    |
| (2, 1)     | 2014           |
| (2, 2)     | [Empty Cell]   |

---

### TODO / PENDING
- Write 2014 revenue (4983791) to (2,2) in top2.xlsx.
- Write 2015 year (3,1) and revenue (4611210) (3,2) in top2.xlsx.
- Generate reports.docx summarizing top 2 revenue years.
- Re-read source rows before writing each numeric value for consistency.

---
```

```
### COMPLETED
- Located and read company_revenues.xlsx.
- Created top2.xlsx.
- Wrote headers and 2014 year to top2.xlsx.
```

## Compression D.4: AppWorld Observation Compression Example

```
task: I am going for a 15-minute drive without internet. Play an album from my
    Spotify library that already has enough downloaded songs for it, so I do not
    have to repeat.

action: print(apis.api_docs.show_api_descriptions(app_name='spotify'))

---------------------------------------

Raw Observation:
[
 {
  "name": "show_account",
  "description": "Show your account information. Unlike show_profile, this includes
    private information."
 },
 {
  "name": "signup",
  "description": "Sign up to create account."
 },
 {
  "name": "delete_account",
  "description": "Delete your account."
 },
 {
  "name": "update_account_name",
  "description": "Update your first or last name in the account profile."
 },
 {
  "name": "login",
  "description": "Login to your account."
 },
 {
  "name": "logout",
  "description": "Logout from your account."
 },
 {
  "name": "send_verification_code",
  "description": "Send account verification code to your email address."
 },
 {
  "name": "verify_account",
  "description": "Verify your account using the verification code sent to your
    email address."
 },
 {
  "name": "send_password_reset_code",
  "description": "Send password reset code to your email address."
 },
 {
  "name": "reset_password",
  "description": "Reset your password using the password reset code sent to your
    email address."
 },
 {
```

```
 "name": "show_profile",
 "description": "Show public profile information of a user."
},
{
 "name": "show_genres",
 "description": "Show the list of all music genres."
},
{
 "name": "search_songs",
 "description": "Search for songs with a query."
},
{
 "name": "show_song",
 "description": "Get details of a specific song."
},
{
 "name": "show_song_privates",
 "description": "Show information about the song that is private to the user."
},
{
 "name": "like_song",
 "description": "Like a song."
},
{
 "name": "unlike_song",
 "description": "Unlike a song."
},
{
 "name": "show_liked_songs",
 "description": "Get a list of songs you have liked."
},
{
 "name": "search_albums",
 "description": "Search for albums with a query."
},
{
 "name": "show_album",
 "description": "Get details of a specific album."
},
{
 "name": "show_album_privates",
 "description": "Show information about the album that is private to the user."
},
{
 "name": "like_album",
 "description": "Like a album."
},
{
 "name": "unlike_album",
 "description": "Unlike a album."
},
{
 "name": "show_liked_albums",
 "description": "Get a list of albums you have liked."
},
{
 "name": "show_playlist_library",
 "description": "Get a list of playlists in the user's playlist library."
},
{
 "name": "search_playlists",
 "description": "Search for playlists with a query. It will search over all public
    playlists and your own private playlists."
```

```
 },
 {
  "name": "create_playlist",
  "description": "Create a new playlist."
 },
 {
  "name": "show_playlist",
  "description": "Get detailed information about a specific playlist. You can view
    your own playlists or others' playlists if they are public."
 },
 {
  "name": "delete_playlist",
  "description": "Delete a playlist."
 },
 {
  "name": "update_playlist",
  "description": "Update a playlist title or privacy."
 },
 {
  "name": "show_playlist_privates",
  "description": "Show information about the playlist that is private to the user."
 },
 {
  "name": "like_playlist",
  "description": "Like a playlist."
 },
 {
  "name": "unlike_playlist",
  "description": "Unlike a playlist."
 },
 {
  "name": "show_liked_playlists",
  "description": "Get a list of playlists you have liked."
 },
 {
  "name": "search_artists",
  "description": "Search for artists with a query."
 },
 {
  "name": "show_artist",
  "description": "Get details of a specific artist."
 },
 {
  "name": "show_artist_following",
  "description": "Show if the user is following the artist."
 },
 {
  "name": "show_song_library",
  "description": "Get a list of songs in the user's song library."
 },
 {
  "name": "add_song_to_library",
  "description": "Add a song to the user's song library."
 },
 {
  "name": "remove_song_from_library",
  "description": "Remove a song from the user's song library."
 },
 {
  "name": "show_album_library",
  "description": "Get a list of albums in the user's album library."
 },
 {
```

```
 "name": "add_album_to_library",
 "description": "Add an album to the user's album library."
},
{
 "name": "remove_album_from_library",
 "description": "Remove an album from the user's album library."
},
{
 "name": "add_song_to_playlist",
 "description": "Add a song to a playlist."
},
{
 "name": "remove_song_from_playlist",
 "description": "Remove a song from a playlist."
},
{
 "name": "show_downloaded_songs",
 "description": "Get a list of downloaded songs."
},
{
 "name": "download_song",
 "description": "Download a song."
},
{
 "name": "remove_downloaded_song",
 "description": "Remove a song from downloads."
},
{
 "name": "show_following_artists",
 "description": "Get a list of artists the user is following."
},
{
 "name": "follow_artist",
 "description": "Follow an artist."
},
{
 "name": "unfollow_artist",
 "description": "Unfollow an artist."
},
{
 "name": "show_song_reviews",
 "description": "Get a list of reviews for a song."
},
{
 "name": "review_song",
 "description": "Rate or review a song."
},
{
 "name": "show_song_review",
 "description": "Show a song review."
},
{
 "name": "delete_song_review",
 "description": "Delete a song review."
},
{
 "name": "update_song_review",
 "description": "Update a song review."
},
{
 "name": "show_album_reviews",
 "description": "Get a list of reviews for an album."
},
```

```
{
 "name": "review_album",
 "description": "Rate or review an album."
},
{
 "name": "show_album_review",
 "description": "Show an album review."
},
{
 "name": "delete_album_review",
 "description": "Delete an album review."
},
{
 "name": "update_album_review",
 "description": "Update an album review."
},
{
 "name": "show_playlist_reviews",
 "description": "Show a list of reviews for your playlist or others' public
   playlist."
},
{
 "name": "review_playlist",
 "description": "Rate or review a playlist."
},
{
 "name": "show_playlist_review",
 "description": "Show a playlist review."
},
{
 "name": "delete_playlist_review",
 "description": "Delete a playlist review."
},
{
 "name": "update_playlist_review",
 "description": "Update a playlist review."
},
{
 "name": "show_payment_cards",
 "description": "Get a list of users payment cards."
},
{
 "name": "add_payment_card",
 "description": "Add a new payment card."
},
{
 "name": "show_payment_card",
 "description": "Get details of a payment card."
},
{
 "name": "delete_payment_card",
 "description": "Delete payment card information."
},
{
 "name": "update_payment_card",
 "description": "Update payment card information."
},
{
 "name": "show_current_song",
 "description": "Show details of the current song on the queue."
},
{
 "name": "play_music",
```

```
  "description": "Play music based on various criteria. You can pass, at most, any
    one of queue_position, song_id, album_id or playlist_id. If one of song_id,
    album_id or playlist_id is passed, that song, album or playlist will be added
    to the queue and played. Otherwise, the queue will remain unchanged. If
    queue_position is passed, the song at that position in the queue will be played
    . If none is passed, the current song in the queue will be played."
},
{
 "name": "pause_music",
 "description": "Pause the currently playing song."
},
{
 "name": "previous_song",
 "description": "Go to the previous song in the song queue."
},
{
 "name": "next_song",
 "description": "Go to the next song in the song queue."
},
{
 "name": "move_song_in_queue",
 "description": "Move a song in the queue to a new position."
},
{
 "name": "seek_song",
 "description": "Seek the current song to the given number of seconds."
},
{
 "name": "loop_song",
 "description": "Set whether to loop the current song."
},
{
 "name": "shuffle_song_queue",
 "description": "Shuffle songs in the music player queue."
},
{
 "name": "show_song_queue",
 "description": "Get the music player song queue. Songs are played in the order of
    the queue in a cycle."
},
{
 "name": "add_to_queue",
 "description": "Add a song, album or playlist to the music player song queue."
},
{
 "name": "clear_song_queue",
 "description": "Clear the music player song queue."
},
{
 "name": "remove_song_from_queue",
 "description": "Remove a song at the given position from the music player song
   queue."
},
{
 "name": "show_volume",
 "description": "Get the volume level of the music player."
},
{
 "name": "set_volume",
 "description": "Set the volume level of the music player."
},
{
 "name": "show_recommendations",
```

```
 "description": "Get personalized song recommendations for the user."
 },
 {
  "name": "show_premium_plans",
  "description": "Show information about premium plans available."
 },
 {
  "name": "show_premium_subscriptions",
  "description": "Show your premium subscription history."
 },
 {
  "name": "subscribe_premium",
  "description": "Subscribe to premium membership."
 },
 {
  "name": "download_premium_subscription_receipt",
  "description": "Download the receipt for a premium subscription."
 }
]

----------------------------------------

Obs. Compression (Prompting baseline):

The Spotify API provides:
- show_album_library: get user's album library.
- show_downloaded_songs: get list of downloaded songs.
- show_album: get details of a specific album.

----------------------------------------

Obs. Compression (ACON (utility step)):
[
  {
    "name": "show_album_library",
    "description": "Get a list of albums in the user's album library."
  },
  {
    "name": "show_downloaded_songs",
    "description": "Get a list of downloaded songs."
  },
  {
    "name": "show_album",
    "description": "Get details of a specific album."
  },
  {
    "name": "play_music",
    "description": "Play music based on various criteria. You can pass, at most,
    any one of queue_position, song_id, album_id or playlist_id. If one of song_id,
     album_id or playlist_id is passed, that song, album or playlist will be added
    to the queue and played. Otherwise, the queue will remain unchanged. If
    queue_position is passed, the song at that position in the queue will be played
    . If none is passed, the current song in the queue will be played."
  }
]

----------------------------------------

History Compression (ACON (utility step + compression step)):

[{"name":"show_album_library","description":"Get user's album library."},{"name":"
    show_downloaded_songs","description":"Get downloaded songs."},{"name":"
```

```
show_album_privates","description":"Show album private info."},{"name":"
play_music","description":"Play music; album_id allowed."}]
```

