# OpenReview forum: "ACON: Optimizing Context Compression for Long-horizon LLM Agents"
_ICML.cc/2026/Conference — ICML 2026 regular_

### Official Review · Reviewer_f8k2 · 2026-03-10

**Soundness:** 3
**Presentation:** 4
**Significance:** 3
**Originality:** 3
**Overall Recommendation:** 4
**Confidence:** 3

**Summary:**

This paper studies the context compression task for agents performing long-horizon tasks. The authors proposes ACON, a method that heuristically finds optimal guidelines for context compression. It achieves this by (1) contrasting successful uncompressed trajectories with failed compressed ones, and (2) extracting clues from successful compressed trajectories. The authors also show that distilling the teacher compressor's capabilities into smaller, deployable models yields promising results, demonstrating the broader applicability of the ACON pipeline.

**Compliance With Llm Reviewing Policy:**

Affirmed.

**Key Questions For Authors:**

Please refer to Weaknesses

**Limitations:**

Yes

**Strengths And Weaknesses:**

### **Strengths**

* ***S1*** **The overall method is intuitive.** While optimizing the compressor is a relatively straightforward technical solution, decoupling the policy model (the agent itself) from the compressor model is an important design choice to maintain their distinct roles. Since evaluating the quality of context compression remains an open problem, using contrastive feedback from both successful and failed trajectories is a clever way to generate an optimal prompt. Besides, I think this dual prompt optimization, first contrasting successes and failures, then distilling clues from successful compressed trajectories, serves as a simple but effective approximation of the overall objective in Equation 7.
* ***S2*** The writing is highly accessible. The clear color coding, vivid figures, and highlighted empirical results make it very easy to grasp the main conclusions quickly.
* ***S3*** The empirical results are extensive. The authors cover diverse benchmarks, different agent models, and evaluate both off-the-shelf and distilled compressors, backing their claims with both quantitative metrics and qualitative case studies.

---

### **Weaknesses**

* ***W1*** Lack of discussion on parallel works like ReSum. I am not criticizing the authors for missing the concurrent work ReSum, but I found that they share the same overall motivation while differing in concrete practice and downstream evaluation. I think adding a discussion comparing the two would improve the depth of the paper. Specifically:
     *  Similar motivation: Both use a model-agnostic summarization module to compress accumulated history and extend the context window. Both are applicable to open-weight and proprietary models, boosting performance even under training-free settings.
     *  Same distillation: Both works distill a specific summary/compressor model to maintain performance while minimizing deployment costs.
     *  Main difference: ReSum optimizes the policy model to act more intelligently when utilizing the compressed context, whereas ACON focuses on quality control of the context itself (specifically the prompt optimization and distillation of the compressor). I think both parts are important: the compression model is essential because it directly impacts whether critical information is lost, while the policy model needs to be aligned to reason well over that summarized context.

* ***W2*** Limited evaluation and motivation regarding context length. Even though benchmarks like AppWorld require 10+ tool calls, the peak token count still easily fits into modern LLM context windows, which now routinely support up to 128K tokens. In these settings, the context length is arguably negligible, which makes it hard to justify the extra API costs and latency of invoking a compressor instead of just running standard ReAct on SOTA LLMs. I think the evaluated benchmarks are not truly challenging "long-horizon" tasks. Evaluating on much longer tasks, such as Deep Research-style information seeking that actually exceeds a 64K or 128K context window, would make the motivation far more convincing.

---

> ### Author Rebuttal · Authors · 2026-03-31
>
> We sincerely thank the reviewer for their valuable and insightful feedback. We are glad to hear that you found our work (1) offers an intuitive and clever dual prompt optimization that effectively decouples the agent and compressor, (2) features highly accessible writing and presentation, and (3) provides extensive empirical results backed by both quantitative metrics and qualitative case studies.
>
> To address your concerns,
> - We clarify how ACON’s core contribution automated guideline optimization prior to distillation differs from and strongly complements ReSum's policy-side RL approach.
> - We acknowledge the limitation regarding ultra-long horizon evaluations for deep research (64K-128K tokens) due to budget constraints, and we provide alternative validations, including new experiments on WebVoyager.
>
> ---
>
> > W1. Lack of discussion on parallel works like ReSum.
>
> We appreciate the reviewer for highlighting ReSum, a highly relevant concurrent work. The core distinction and synergy between our approach are as follows:
>
> - While both methods decouple the compressor and utilize distillation, ReSum optimizes the policy model via RL to utilize compressed contexts, whereas **ACON focuses on the quality control of the compressed context itself.** ACON actively learns optimal, environment-specific compression rules via contrastive feedback prior to distillation, ensuring the distilled compressor learns from a highly refined policy rather than generic summaries.
> - **ACON's natural language optimization can be directly applied even when the setup relies purely on black-box, proprietary models.** Because this optimization happens purely in the prompt space, it is extremely lightweight (costing under $2) and allows practitioners to easily adapt to highly diverse domains.
> - We view ReSum’s approach as a huge contribution to the field, and **ACON operates complementarily to such methods.** ACON provides high-fidelity, environment-optimized compressed context, while a ReSum-like RL approach can align the policy model to reason optimally over it.
>
> We have added a detailed comparison in `Related Works` and `Appendix C` of the revision.
>
> ---
>
> > W2. Limited evaluation and motivation regarding context length.
>
> We agree with the reviewer that evaluating on ultra-long horizon tasks like Deep Research is the ideal setting to demonstrate the limits of context management.
>
> While full-scale rollouts for such tasks were restricted by high API/Search engine costs, we have provided extensive evidence of ACON’s utility in high-pressure contexts:
> - To further validate generalizability, we additionally conducted **new experiments on the web agents task** (WebVoyager benchmark). ACON outperforms no compression by +12.9 in accuracy while reducing peak token by ~40%.  Detailed results are in our response to `Reviewer gwmM W2&Q1`.
> - **Even within 10K-20K contexts, smaller models suffer from context distraction.** ACON bridges the performance gap for these models, boosting Qwen3-14B's performance by 45.6% on 8-objective QA. This demonstrates that ACON’s benefits in memory efficiency and reasoning quality are important for making smaller, deployable models viable in long-horizon tasks.

---

> > ### Author Rebuttal · Reviewer_f8k2 · 2026-04-03
> >
> > Thank you for the response. I will maintain my positive score and continue to champion this paper.

---

> > > ### Author Response · Authors · 2026-04-08
> > >
> > > Dear Reviewer f8k2,
> > >
> > > Thank you for your final response and for your strong support of our work. We are truly encouraged by your decision to continue championing our paper. We are also glad that **our rebuttal fully addressed your concerns.**
> > >
> > > We appreciate the time and effort you invested in providing such constructive and insightful feedback.
> > >
> > > Best regards,
> > >
> > > The Authors

---

### Official Review · Reviewer_gwmM · 2026-03-12

**Soundness:** 3
**Presentation:** 3
**Significance:** 2
**Originality:** 2
**Overall Recommendation:** 4
**Confidence:** 4

**Summary:**

This paper proposes a technique for compressing history / states for agents via prompt optimization. Concretely, the method collects trajectories with and without compression, and perform prompt optimization via (1) comparing successful trajectory without compression with failed trajectory with compression and (2) reducing length for successful compression.

The paper experiments with using gpt-4.1 as both the agent and compressor and evaluated on Appworld, OfficeBench and 8-objective QA, showing varying performance compared to no compression / other baselines. Distilling the compressor into a smaller model (e.g. QWEN-14B) show slightly degraded performance.

**Compliance With Llm Reviewing Policy:**

Affirmed.

**Key Questions For Authors:**

* What are the performance of the proposed method on other agents and benchmarks? (WebArena)
* How does applying more prompt optimization steps impact the results?
* What is the end-to-end latency for the proposed method if the main advantage of the proposed method is efficiency?
* The paper separated history compression and observation compression - what's the motivation for the design choice and what would happen if we compress both of them?

**Limitations:**

Yes

**Strengths And Weaknesses:**

**Strength**
* The paper studies a well-motivated problem: context compression for long-horizon agents.
* The paper is written clearly.


**Weakness**
* While the motivation is clear, the main advantage for the method is a bit unclear to me. If the idea is to reduce the inference time for the agent, reporting token counts is insufficient as the compressor themselves will also take up some time. Reporting the end-to-end efficiency would be good. If the idea is to improve performance by allowing for longer interaction, I don't think the current set of experiments support the claim as performance degrades for OfficeBench and 8-objective QA compared to no compression.
* Limited set of benchmark and agent model evaluated: one of the advantage of the proposed method is that it is "model-agnostic", it would be helpful to evaluate the method for other agents aside from gpt-4.1. The 8-objective QA task is not completely clear and well-motivated for an "agent" task; evaluating on more well-studied benchmark such as WebArena would make it easier for the community to interpret the result.
* Overall I find the claim of "model optimization-based approach inapplicable to proprietary, API-based LLM" misleading - one can train the compressor model by collecting reward from the  API-based model, similar to the proposed method. A better comparison with parameter update approach should be included (e.g. learning efficiency, reward signal).

---

> ### Author Rebuttal · Authors · 2026-03-31
>
> We sincerely thank the reviewer for their valuable and insightful feedback.
> We appreciate that the reviewer recognized (1) our work studies a well-motivated problem of context compression for long-horizon agents and (2) the paper is written clearly.
>
> To address your concerns,
> - We clarify that ACON majorly targets memory efficiency and improves performance compared to compression baselines.
> - We include new experimental results on WebVoyager.
> - We distinguish ACON from parameter-update methods by highlighting its applicability to closed API-based models.
> - We justify our design choices for a single optimization round and separated compression.
>
> ---
>
> > W1 & Q3. Efficiency and End-to-End Latency & Performance Degradation
>
> Thank you for the careful feedback. We respectfully clarify that **the main advantage of ACON is achieving superior memory efficiency while outperforming existing compression baselines**.
> - From a deployment perspective, memory efficiency directly reduces the footprint per agent. This enables **a higher density of agent deployments within a fixed computational budget**, which is important for system reliability and scaling.
> - While a generative compressor introduces additional latency, an inherent trade-off in current compression paradigms (`Appendix A`), the gains are substantial. As detailed in our response to `Reviewer pgxM W3` and `Q1`, this modest latency increment enables a **26-54% reduction in memory cost and 14% reduction in end-to-end costs.**
> - **Furthermore, ACON improves the performance of smaller models.** As highlighted in `Figure 5`, smaller models like Qwen3-14B often benefit from compressed contexts from ACON, achieving a 32.4% relative performance improvement on AppWorld.
>
> We have updated the revision to clarify this distinction, modifying “preserving task success” to “improving task success compared to compression baselines”. While slightly behind 'no compression' in OfficeBench, ACON offers a better performance-efficiency trade-off than existing baselines.
>
> ---
>
> > W2 & Q1. Limited set of benchmark and agent model evaluated. What is the performance of the proposed method on other agents and benchmarks?
>
> We agree that evaluating on diverse models and well-studied benchmarks strengthens the paper. We would like to clarify that we already evaluated ACON on multiple models, and we have added new results on web agent benchmark (WebVoyager) as requested.
> - **Diverse Models:** We already evaluated ACON on a wide range of models beyond gpt-4.1. As shown in `Appendix C.1` and `Section 4.4`, we included results that apply ACON to different API and open-weight models like gpt-4.1-mini, gpt-5-chat, and Qwen3-14B.
> - **Why 8-Objective QA:** We utilized 8-objective QA because it simulates a deep research style agentic task following [1], which does not require the use of expensive commercial search engines.
> - **WebVoyager experiment:** Following your valuable suggestion, we are conducting additional experiments on WebVoyager. As in the table below, **ACON outperforms no compression while reducing peak token and dependency**, validating its efficacy in web-agent tasks.
> |   |acc|peak|dep|
> |---|---:|---:|---:|
> |No comp|35.7|1.3k|2.53M|
> |Hist prompting|42.9|8k|1.23M|
> |Hist ACON|48.6|8k|1.19M|
> |Obs prompting|45.7|4k|0.85M|
> |Obs ACON|47.1|4k|0.8M|
>
> We will include experimental details and results on WebVoyager in the revision.
>
> [1] Zhou et al., MEM1: learning to synergize memory and reasoning for efficient long-horizon agents
>
> ---
>
> > W3. The misleading claim
>
> We respectfully clarify that our claim in Section 1 refers to the technical limitation that **weight of proprietary API-based LLM  cannot be directly updated** (e.g., using gpt-4.1 as the compressor). This highlights our optimization method is applicable to both proprietary and open-weight models, unlike prior weight-update methods.
>
> To ensure better precision, we have reworded "inapplicable to proprietary model" to "difficult to apply to proprietary model as gradient updates to these models are infeasible" in Section 1 of the revision.
>
> ---
>
> > Q2. More prompt optimization steps
>
> **We included the additional optimization step experiment in `Appendix C.2 (Table 12)`** and found that a single round of optimization is sufficient.
> - The results show that this additional iteration (UT $\rightarrow$ CO $\rightarrow$ UT) results in a performance drop.
> - This indicates that our proposed single round of optimization is already sufficient for effective guideline optimization, preventing overfitting to the training set.
>
> ---
>
> > Q4. Separated history and observation compression
>
> Our motivation was to demonstrate that our unified optimization framework effectively optimizes both compression paradigms.
> - We can use both simultaneously, and **we already provided this exact experiment in `Appendix C.2 (Table 12)`**.
> - As expected, combining them achieves larger reductions in peak token usage and dependency but more performance drop.

---

> > ### Author Rebuttal · Reviewer_gwmM · 2026-03-31
> >
> > Thank you for the response and new results on WebVoyager! While it is unclear which model this was tested on, the performance improvement is encouraging, especially if it is tested on larger model such as gpt-4.1.
> > While I still have some concern for the paper, I have increased my score to 4 given the clarification on improvement for smaller models. It would be great if the authors can discussed it in the manuscript:
> > * The main motivations for the proposed method, is it for efficiency purpose or for performance improvement. While the proposed method benefits small models, they do lead to performance degradation for gpt-4.1 (table 2). If it is for efficiency purpose, I think the definition / measurement for efficiency should be clarified and comprehensively considered.

---

> > > ### Author Response · Authors · 2026-04-01
> > >
> > > Thank you for engaging so deeply with our work. We greatly appreciate your insightful suggestions, your recognition of the performance improvements for smaller models, and the increased score. We address your follow-up points as follows:
> > >
> > > > 1. WebVoyager Model
> > >
> > > The WebVoyager experiments were indeed evaluated using **gpt-4.1**. We apologize for the omission and will include these details in the revision.
> > >
> > > > 2. Clarifying the Motivation
> > >
> > > You are completely right to point out the performance degradation for gpt-4.1 in Table 2. We will comprehensively clarify in the revision that our main motivation is memory efficiency via optimized context compression, which yields better performance than existing compression baselines.
> > >
> > > Our core motivation is formulated in **Equation 7**, which explicitly models the trade-off between maximizing task success (reward) and minimizing memory (context) cost. To explain this further:
> > > - **The Problem:** Existing baseline compression methods aggressively minimize memory cost but can cause severe performance drops compared to *no compression* since critical information can be lost.
> > > - **Our Solution (ACON):** By optimizing the compression process, ACON effectively solves the objective in Equation 7, matching the memory efficiency gains of the *compression baselines* while significantly improving performance over them.
> > > - **The Trade-off:** As you correctly noted, while ACON achieves *no compression* level performance on certain benchmarks (like AppWorld), there is still some performance degradation on others for gpt-4.1 (Table 2).
> > >
> > > > 3. Manuscript Updates on Claims and Efficiency
> > >
> > > To avoid any misleading claims regarding performance, we will update the text to explicitly state that ACON shows better performance **compared to existing compression baselines**, rather than an absolute performance increase for large models compared to no compression.
> > > For instance, we will update the claim (Lines 104-109):
> > > - From: "...reduces peak token usage by 26-54% while preserving task success with large LLMs"
> > > - To: "...reduces peak token usage by 26-54% while **improving task success compared to existing compression baselines** with large LLMs."
> > >
> > > Additionally, we will add a dedicated section to comprehensively define and discuss our measurements for efficiency (token usage, end-to-end costs, and the latency trade-offs).
> > >
> > > Thank you again for your valuable time and effort in helping us make the paper's claims much more precise and well-grounded. If you have any remaining concerns or questions, please let us know. We are happy to discuss them further.
> > >
> > > Best regards,
> > >
> > > The Authors

---

### Official Review · Reviewer_pgxM · 2026-03-12

**Soundness:** 3
**Presentation:** 3
**Significance:** 3
**Originality:** 3
**Overall Recommendation:** 4
**Confidence:** 4

**Summary:**

This paper presents ACON, a unified framework for context compression in long-horizon LLM agent tasks. To address the accumulation of multi-step interaction histories and verbose environment observations, ACON analyzes contrastive trajectory pairs—where full-context execution succeeds but compressed-context execution fails—to identify failure causes via an LLM and iteratively refine compression rules in natural language. Experiments on three benchmarks (AppWorld, OfficeBench, Multi-objective QA) demonstrate that ACON reduces peak token usage by 26–54% while maintaining task accuracy comparable to or exceeding uncompressed baselines.

**Compliance With Llm Reviewing Policy:**

Affirmed.

**Final Justification:**

As mentioned in the rebuttal acknowledgement. I have no further concerns and decide to keep the positive rating.

**Key Questions For Authors:**

Q1. Could the authors supplement the evaluation with inference latency or wall-clock time metrics, and provide an estimated end-to-end cost (e.g., API token cost) for running a full ACON pipeline on AppWorld, compared against the primary baselines?

Q2. Does ACON include any explicit mechanism to ensure that the compressor LLM strictly consolidates past interaction history without inadvertently introducing additional reasoning steps or implicit hints that could artificially inflate the downstream agent's task performance?

Q3. How does the compression rule set generalize to entirely unseen task domains? Could the authors provide even preliminary evidence—such as zero-shot transfer to a held-out domain—to address the potential overfitting concern?

**Limitations:**

yes

**Strengths And Weaknesses:**

Strengths

1. The paper targets a practical and underexplored bottleneck in long-horizon agentic settings: the compounding memory and inference cost arising from accumulated multi-step interaction histories and environment observations.

2. The trajectory-contrastive rule refinement mechanism operates entirely in natural language, making the compression policy human-readable and avoiding the need for gradient-based training. The self-reflective rule update loop is a technically clean design with good interpretability.

3. Results in Tables 1 and 2 demonstrate that ACON achieves task accuracy close to or exceeding uncompressed full-context baselines across multiple benchmarks.

Weaknesses

1. The primary baselines are full-context execution and simple truncation or summarization approaches. However, no comparison was made with the more recent context compression methods proposed after 2024.

2. The paper demonstrates strong performance in in-distribution settings, but does not investigate whether the compression rules distilled into the smaller compressor model overfit to the specific domains used during training. It remains unclear whether ACON's compressor would retain its effectiveness—without dropping critical information—when applied to task domains not encountered during the rule refinement phase.

3. ACON introduces an additional LLM call at each compression step. While the paper reports Peak and Dep. token metrics following the convention of LightThinker, it does not report detailed wall-clock inference time or end-to-end latency.

---

> ### Author Rebuttal · Authors · 2026-03-31
>
> We sincerely thank the reviewer for their valuable feedback. We are glad to hear that you found our work (1) targets a practical and underexplored bottleneck, (2) offers a technically clean and interpretable design, and (3) achieves strong performance across multiple benchmarks.
> To address your concerns,
> - We clarify the differences between ACON and recent compression methods.
> - We address the generalization concern by explaining that train/test tasks are thoroughly distinct.
> - We report empirical latency and API cost data, showing that ACON reduces cost by ~15% with modest latency overhead.
>
> ---
>
> > W1. Baselines
>
> We deeply appreciate your suggestion regarding baselines. We would like to clarify that comparisons with agent-specific methods (e.g., MEM1, AgentFold) are discussed in **`Section 2` and `Appendix C.3 (Table 5)`.**
> - Most of recent compression methods rely on joint reinforcement learning of the agent and the compression policy. This inherently requires updating the agent model’s weights.
> - In contrast, ACON uses decoupled optimization in natural language space for compressors. This design offers unique advantages: (1) no model weight training, (2) support for proprietary APIs (e.g., gpt-4.1), and (3) a Large Agent + Small Compressor architecture.
> - Due to these different constraints (weight-access vs. API-based), a direct empirical comparison is not straightforward.
>
> ---
>
> > W2 & Q3. Overfitting of the compressor: How does the compression rule set generalize to entirely unseen task domains?
>
> We would like to clarify that **ACON is intentionally designed to learn environment-specific compression rules, rather than a single universal rule.**
> - Our baseline experiments (Naive Prompting) demonstrate that a universal compression instruction often fails to retain important information, leading to severe performance drops.
> - Therefore, ACON does not aim for zero-shot transfer of a single rule across entirely unseen domains. Instead, **the generalization capability of ACON lies in its optimization pipeline**, which can automatically derive an optimized, domain-specific guideline for any new domain at a negligible cost (under $2).
> - We emphasize that within our benchmarks, **train tasks and test tasks are explicitly distinct** and our optimized compressor maintains strong performance on the unseen test set.
>
> ---
>
> > W3.  Wall-clock inference time or end-to-end latency.
>
> While ACON introduces an additional LLM call for compression, latency increase is relatively modest. The median end-to-end latency per task was 73.24s (No Comp), 87.68s (History Comp), and 101.92s (Obs Comp) on AppWorld (Qwen3-14B).
>
> This is a justifiable trade-off for the following reasons:
> - **ACON significantly reduces the memory cost** such as peak tokens and dependency, preventing the GPU memory explosion that can occur in uncompressed long-horizon tasks.
> - In the Qwen3-14B model, the **optimized compressions of ACON can contribute to improve the performance** as shown in `Figure 5`.
> - We acknowledge that introducing an LLM for compression adds generative latency, which is **a known fundamental bottleneck shared by all generative LLM-based compression methods**. We openly discussed this limitation in `Appendix A`.
>
> We have included discussion on the performance-latency trade-off in the revision.
>
> ---
>
> > Q1. API token cost
>
> We provide the estimated end-to-end API token cost on AppWorld, demonstrating that **ACON effectively reduces the total operational cost compared to most baselines.**
>
> | Appworld              | Total API Cost |
> |---|---:|
> | No Compression | 0.331 |
> | History compression |      |
> | FIFO | 0.390       |
> | LLMLingua | 0.392  |
> | ACON | 0.285  |
> | Observation Compression  |    |
> | LLMLingua  | 0.331   |
> | ACON  | 0.272 |
> *per task cost using gpt-4.1 agent, qwen3-14B compressor*
>
> - Unlike simpler baselines like FIFO or LLMLingua, ACON avoids step inflation by preserving critical task information, leading to lower total agent costs.
> - **ACON achieves a ~14% lower total system cost than the no compression baseline**, proving that optimized compression can be more efficient as reduced context length lowers the agent's tokens.
>
> ---
>
> > Q2. Performance Inflation
>
> Yes, ACON explicitly prevents the compressor from injecting artificial reasoning or hints through both the dynamic nature of the tasks and strict structural constraints in guidelines.
> - In dynamic agent environments, **essential task information is hidden and must be retrieved via agent actions.** Thus, the compressor cannot hallucinate reasoning paths or answers in advance as it lacks access to the dynamic environment state.
> - As detailed in `Prompts D.6` and `D.7`, the optimized guidelines enforce a rigid output schema. The compressor is explicitly instructed: "Do not output any commentary" and "Never invent or alter state variables.". This strictly bounds the compressor to extraction and factual summarization, removing the risk of artificial hinting.

---

> > ### Author Rebuttal · Reviewer_pgxM · 2026-04-03
> >
> > Thanks for the authors' response. I have no further questions and will keep the positive evaluation.

---

> > > ### Author Response · Authors · 2026-04-08
> > >
> > > Dear Reviewer pgxM,
> > >
> > > Thank you for confirming that **your concerns are fully resolved**. We are pleased to hear that our rebuttal clarified the points regarding baselines, generalization, and costs.
> > >
> > > We truly appreciate your support and the positive evaluation of our work.
> > >
> > > Best regards,
> > >
> > > The Authors

---

### Official Review · Reviewer_DQa6 · 2026-03-13

**Soundness:** 4
**Presentation:** 4
**Significance:** 3
**Originality:** 3
**Overall Recommendation:** 5
**Confidence:** 4

**Summary:**

This paper proposes ACON, a framework for optimizing context compression in long-horizon LLM agents by learning compression guidelines that condense interaction histories and environment observations while preserving task-relevant information. The method iteratively refines (i.e., in natural language) compression guidelines using failure analysis and further reduces overhead by distilling the optimized compressor into smaller models, as well as by showing how smaller black-box models can still be effective compressors once guidelines are refined. Experiments on three long-horizon benchmarks show substantial reductions in memory usage with minimal performance loss, and in some cases improved effectiveness for smaller models.

**Compliance With Llm Reviewing Policy:**

Affirmed.

**Key Questions For Authors:**

Please refer to Weaknesses.

**Limitations:**

Yes.

**Strengths And Weaknesses:**

Strengths:

1. The paper is well-written and well-organized.

2. The problem of context compression is well-motivated and relevant, and solving it by iteratively refining compression guidelines makes a lot of sense, especially in keeping the solution applicable to black-box models.

3. The experimental set up is well-designed: three benchmarks, three well-justified compression metrics, with sensible baselines across the experimental space. Analyses are also meaningful, especially how the compressor can be downsized once the guidelines have been optimized, either through distillation or using a mini pretrained model.

4. Overall, I think this paper makes relevant contributions to the research community studying long-horizon agents.

Weaknesses:

1. The main ambiguity from the paper's results lies on whether (and when) to use ACON-UT or ACON-UTCO: for AppWorld, ACON-UTCO outperforms ACON-UT, whereas it's the opposite for OfficeBench and 8-objective QA. The authors should better discuss this fluctuation in relative performance, providing further insight into how to decide between the two variants. Although well-written as is, Section 3 could be streamlined (i.e., all the preparation before "Optimization objective") to make room for this additional discussion.

---

> ### Author Rebuttal · Authors · 2026-03-31
>
> We sincerely thank the reviewer for their valuable and thoughtful feedback.
> We are glad to hear that you found our work (1) addresses a well-motivated and relevant problem for long-horizon agents, (2) features a well-designed experimental setup with sensible baselines, and (3) provides meaningful analysis on downsizing compressors via distillation.
>
> To address your concerns,
> - We clarify the performance fluctuation between ACON-UT and ACON-UTCO and provide a practical guideline on when to use each variant based on environment characteristics.
> - We have incorporated the valuable discussion on UT and CO into Section 3 of the revision.
>
> ---
>
> > W1. Ambiguity of UT & UTCO
>
> We completely agree with the reviewer that discussing the trade-off between UT and UTCO is an important insight for practitioners, and we deeply appreciate this suggestion.
> - **The choice between UT and UTCO fundamentally depends on the verbosity and noise level of the environment's observations.** UT (Utility Maximization) focuses on preserving task-critical information to maximize accuracy, while CO (Compression Maximization) focuses on aggressively stripping redundant structures to minimize context cost.
> - **In highly verbose and noisy environments like AppWorld, UTCO is the preferred choice** because aggressive compression actively filters out distractors such as API logs, improving both efficiency and accuracy. The CO step learns to aggressively prune this excess noise, which actually helps the agent focus on the core reasoning without being distracted, leading to a higher task success rate.
> - **In environments demanding high-fidelity fact retention like OfficeBench and QA, UT is generally preferred to prioritize accuracy**, as over-compression may discard subtle but necessary details. These tasks require precise reading of documents or retrieved text snippets. While applying UTCO achieves more token reduction, this aggressive pruning can occasionally remove nuanced facts, causing a slight drop in the final score.
>
> We have included a thorough discussion of this trend and provide these actionable insights in Section 3 method and 4 experiments of the revision. Your constructive feedback will significantly improve the practical utility and clarity of our paper.

---

> > ### Author Rebuttal · Reviewer_DQa6 · 2026-04-02
> >
> > I thank the authors for their rebuttal and specifically for the additional insights into UT vs. CO, which address my initial comments.

---

> > > ### Author Response · Authors · 2026-04-08
> > >
> > > Dear Reviewer DQa6,
> > >
> > > Thank you for your positive assessment and for confirming that **our rebuttal addressed your concern** regarding UT vs. CO.
> > >
> > > We sincerely appreciate your insightful feedback and the time you dedicated to improving our work throughout the review process.
> > >
> > > Best regards,
> > >
> > > The Authors

---

### Decision · Program_Chairs · 2026-04-30

**Decision:**

Accept (regular)

**Comment:**

ACON addresses the well-motivated problem of context management in long-horizon LLM agents through a practical, model-agnostic compression pipeline. The reviewers agreed on the technical soundness and utility of distilling trajectory-based compression guidelines into smaller models.

While initial concerns focused on efficiency trade-offs, evaluation breadth, and variant distinctions, the rebuttal adequately resolved these by clarifying the paper’s positioning (efficiency with better-than-baseline performance), adding latency/cost breakdowns, and providing extra experimental results.

I am recommending acceptance based on the paper’s clear practical value and solid empirical foundation. It is important to note the paper’s inherent boundaries: the approach is rooted in heuristic prompt optimization rather than a fundamental algorithmic leap, and its advantages over full-context baselines are not absolute for the most capable large models. However, as a pragmatic systems contribution to the agent community, it successfully meets the acceptance threshold.